# Learning When to Think: Shaping Adaptive Reasoning in R1-Style Models via Multi-Stage RL

**Songjun Tu**[1,2,3], **Jiahao Lin**[1,3], **Qichao Zhang**[*1,3], **Xiangyu Tian**[1,3],
**Linjing Li**[‡,3], **Xiangyuan Lan**[2], **Dongbin Zhao**[1,2,3]
[1]Institute of Automation, Chinese Academy of Sciences  [2]Pengcheng Laboratory
[3]School of Artificial Intelligence, University of Chinese Academy of Sciences
{tusongjun2023,zhangqichao2014}@ia.ac.cn

## Abstract

Large reasoning models (LRMs) are proficient at generating explicit, step-by-step reasoning sequences before producing final answers. However, such detailed reasoning can introduce substantial computational overhead and latency, particularly for simple problems. To address this overthinking problem, we explore how to equip LRMs with adaptive thinking capabilities, enabling them to dynamically decide whether to engage in explicit reasoning based on problem complexity. Building on R1-style distilled models, we observe that inserting a simple ellipsis ("...") into the prompt can stochastically trigger either a thinking or no-thinking mode, revealing a latent controllability in the reasoning behavior. Leveraging this property, we propose *AutoThink*, a multi-stage reinforcement learning (RL) framework that progressively optimizes reasoning policies via stage-wise reward shaping. Auto-Think learns to invoke explicit reasoning only when necessary, while defaulting to succinct responses for simpler tasks. Experiments on five mainstream mathematical benchmarks demonstrate that *AutoThink* achieves favorable accuracy–efficiency trade-offs compared to recent prompting and RL-based pruning methods. It can be seamlessly integrated into any R1-style model, including both distilled and further fine-tuned variants. Notably, *AutoThink* improves relative accuracy by 6.4% while reducing token usage by 52% on DeepSeek-R1-Distill-Qwen-1.5B, establishing a scalable and adaptive reasoning paradigm for LRMs.

**Project Page: https://github.com/ScienceOne-AI/AutoThink.**

## 1 Introduction

Recently, reasoning-focused Large Language Models (LLMs), also referred to as Large Reasoning Models (LRMs) [41], have demonstrated remarkable progress in solving complex reasoning tasks. Particularly, DeepSeek-R1 [9] uses only outcome-based feedback and incentivizes explicit reasoning capabilities through reinforcement learning (RL) with verifiable rewards. DeepSeek-R1 and its distilled models typically follow the <think> and <answer> format, where the <think> process generates explicit, step-by-step reasoning sequences to support obtaining a final answer during the <answer> phase. We refer to models that follow this Chain of Thought (CoT) [37] prompting scheme as *R1-style models*. The explicit thinking process, which enables self-reflection, backtracking, and validation, is widely regarded as essential for enhancing reasoning accuracy. Arising from this understanding, a popular paradigm has emerged that improves solution quality by increasing thinking token allocation during inference-time reasoning [17, 40]. However, this paradigm introduces a major bottleneck: excessive thinking token generation leads to high computational cost and latency, raising the *overthinking* phenomenon, where many reasoning steps are redundant or inefficient [29, 14].

---

[*]Corresponding author
[†]Project leader

39th Conference on Neural Information Processing Systems (NeurIPS 2025).

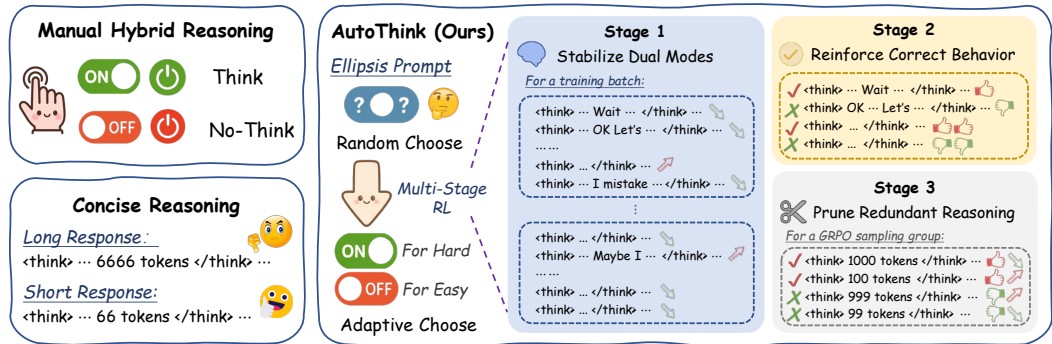

Figure 1: Overview of *AutoThink* Compared to Prior Reasoning Paradigms.

To mitigate overthinking, recent efforts have explored *hybrid reasoning* and *concise reasoning* strategies. In the industry, Claude 3.7 Sonnet [2] introduces a controllable reasoning framework that allows the model to switch between standard and extended reasoning modes. Similarly, Qwen3 [30] proposes a thinking control scheme with a "thinking" mode *(slow thinking)* and a "non-thinking" mode *(fast thinking)*, and provides users with the flexibility to choose whether the model should engage in reasoning or not. In the academic community, parallel research has focused on designing prompt-guided efficient reasoning [42, 24] or training pruning-based models to achieve concise reasoning [12, 6, 44]. While promising, these approaches either rely on manually predefined modes or uniformly prune reasoning steps, which may degrade performance on harder instances. A fundamental question then arises to address the overthinking issue:

> ***Can LLMs learn to adaptively determine thinking fast or slow based on given problems?***

To answer this question, we propose *AutoThink*, a multi-stage RL framework that enables R1-style LLMs to learn adaptive reasoning behaviors. Unlike prior approaches reliant on hard-coded prompting or external control signals, *AutoThink* formulates reasoning as a learned dual-mode policy that determines both whether to engage the model's "thinking" process and how to generate concise reasoning. As illustrated in Figure 1, *AutoThink* fundamentally differs from manual hybrid prompting and uniform pruning strategies by employing an ellipsis prompt and structured three-stage RL training process that enables adaptive reasoning to emerge. In detail, an ellipsis prompt acts as a controllable entry point for optional reasoning, triggering stochastic switching between thinking and no-thinking modes in R1-style LLMs. Then, the proposed multi-stage RL framework shapes this behavior progressively: Stage 1 stabilizes dual-mode coexistence, Stage 2 reinforces accurate reasoning to enhance solution quality, and Stage 3 prunes redundancy via length-aware rewards. This progression enables the model to allocate reasoning effort adaptively, achieving both accuracy and efficiency. The main contributions are as follows:

- We identify the ***ellipsis prompt***, a lightweight prompting scheme that activates a stochastic switching behavior in R1-style LLMs between thinking and no-thinking modes.
- We propose a ***multi-stage RL*** framework that trains R1-style LLMs to dynamically modulate their reasoning behaviors according to problem complexity.
- Experiments on mathematical benchmarks show that *AutoThink* achieves ***accuracy–efficiency trade-offs*** better than existing pruning and compression methods, without sacrificing performance.

## 2 An Ellipsis Unlocks Random Thinking in R1-Style Models

### 2.1 A Surprising Effect of Minimal Prompt Modification

Recent efforts on concise reasoning aim to eliminate unnecessary thought, either via prompting that explicitly bypasses thinking [19], or RL-based training that penalizes long outputs [6]. While effective at shortening responses, these methods enforce uniform brevity regardless of problem complexity. Rather than compressing by default, we pose a subtler question:

> *Can a small change, perhaps a few tokens, lead R1-style models to **decide whether to think**?*

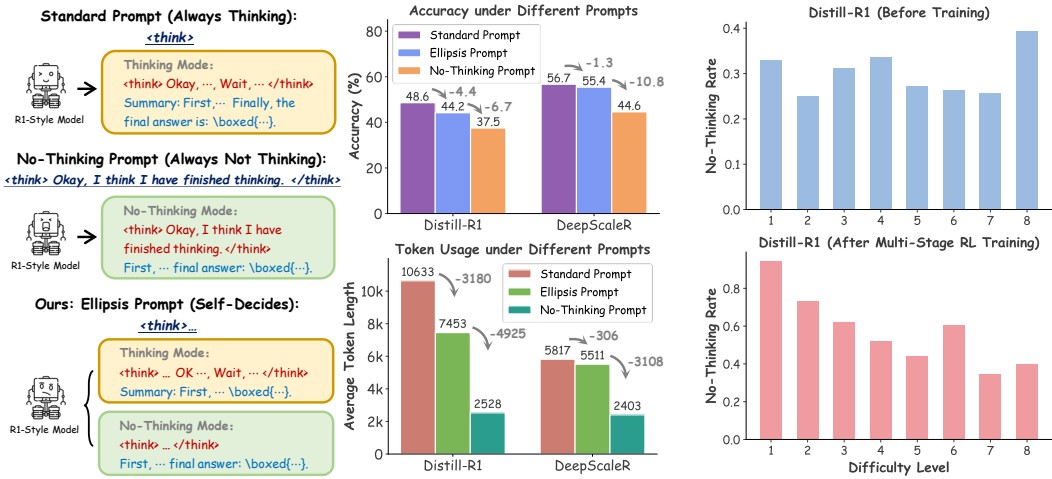

(a) Accuracy and token usage with standard and ellipsis prompts.     (b) No-Thinking rate by difficulty.

Figure 2: Prompting strategies shape reasoning behavior and computational cost.

To investigate this question, we explore how a minimal modification to the prompt structure can influence reasoning behaviors in R1-style models. The baseline prompt used typically includes a `<think>\n` tag followed by a fixed, detailed reasoning trace. In contrast, our modified prompt contains only a single ellipsis following the baseline tag. Specifically, the final prompt we provide is: `<think>\n...\n`. This minimal form acts as an open-ended signal, leaving it entirely up to the model to decide whether to engage in thinking, how much to elaborate, and when to stop.

Surprisingly, this tiny change leads to a distinct shift in behavior. Without any additional training, the model often generates a closing `</think>` tag, sometimes immediately, skipping deep thinking entirely, and other times after producing a full derivation. As shown in Figure 2a, evaluation on Distill-R1-1.5B [9] and DeepScaleR [17] across five mathematical benchmarks shows that ellipsis prompting leads to a modest drop in accuracy, accompanied by a substantial reduction in token usage.

Compared to the *no-thinking prompt* baseline [19], which suppresses reasoning at the cost of accuracy, **the ellipsis prompt triggers a stochastic switch in reasoning mode and provides a more balanced trade-off by preserving reasoning when needed and reducing unnecessary computation**.

## 2.2 Prompting Alone Does Not Enable Difficulty-Aware Thinking

The proposed ellipsis prompt seems to trigger selective reasoning: the model thinks on some inputs but not others. While this behavior appears desirable, it raises a deeper question:

> *Does the prompt-forcing model choose to engage in deep thinking **based on task difficulty**?*

Ideally, a well-calibrated model should reason more on complex problems and skip unnecessary thinking on simpler ones. To assess this, we divide MATH500 problems into 8 difficulty levels based on the average accuracy of Distill-R1 (standard prompt) over 16 rollouts, with higher accuracy indicating lower difficulty. Figure 2b (top) shows the no-thinking rate across these levels. Contrary to expectations, under the ellipsis prompt without additional training, no clear trend emerges—**the flat distribution suggests that thinking is unguided and unaffected by problem complexity.**

A decreasing no-thinking rate along the difficulty axis reflects a desirable reasoning pattern, in which the model allocates effort based on task difficulty. However, this behavior does not emerge from prompting alone. Even with diverse prompt designs (Appendix A.2), the model failed to exhibit difficulty-aware reasoning. Yet prompt-only control suffers from a core limitation: **without feedback, the model lacks a mechanism to learn when the thinking process is needed.**

To address this gap, we introduce a multi-stage RL framework that rewards appropriate reasoning behavior and encourages alignment between effort and difficulty. As shown in Figure 2b (bottom), the resulting distribution from our final trained model exhibits clear difficulty-aware reasoning.

## 3 Guiding When to Think via Multi-Stage Reinforcement Learning

We propose **AutoThink**, a multi-stage RL framework with three training phases that induce difficulty-aware reasoning through progressively refined reward designs. At all stages, we employ the GRPO algorithm with a token-level policy gradient loss [25, 46]. The training objective is:

$$\mathcal{J}_{\text{GRPO}}(\theta) = \mathbb{E}_{(q,a) \sim \mathcal{D}, \{o_i\}_{i=1}^G \sim \pi_{\theta_{\text{old}}}(\cdot|q)} \tag{1}$$
$$\left[ \frac{1}{\sum_{i=1}^G |o_i|} \sum_{i=1}^G \sum_{t=1}^{|o_i|} \min \left( r_{i,t}(\theta) \hat{A}_{i,t}, \text{clip} \left( r_{i,t}(\theta), 1 - \varepsilon, 1 + \varepsilon \right) \hat{A}_{i,t} \right) \right]$$

Here, $o_i$ denotes the $i$-th sampled output for a given query $q$; $G$ is the number of sampled outputs per query; $r_{i,t}(\theta)$ is the token-level importance weight, defined as the ratio between the new and old token probabilities; and $\hat{A}_{i,t}$ represents the estimated token-level advantage. The overall loss is normalized by the total number of tokens across all sampled trajectories. A visual overview of the reward mechanisms across the three training stages is illustrated in Figure 1. In the following subsections, we detail the reward design for each stage.

### 3.1 Stage 1: Preventing Mode Collapse by Batch Reward Balance

To promote efficient reasoning, higher rewards are assigned to correct answers without thinking, and stronger penalties to incorrect ones. Define $\text{think}_i \in \{0, 1\}$ as an indicator of whether the $i$-th output involves thinking, and $\text{correct}_i \in \{0, 1\}$ as an indicator of whether it yields the correct answer. Based on these variables, the naive reward assignment is:

$$r_i^{\text{naive}} = \begin{cases} +1, & \text{if think}_i = 1 \wedge \text{correct}_i = 1, \\ 0, & \text{if think}_i = 1 \wedge \text{correct}_i = 0, \\ +2, & \text{if think}_i = 0 \wedge \text{correct}_i = 1, \\ -1, & \text{if think}_i = 0 \wedge \text{correct}_i = 0. \end{cases} \tag{2}$$

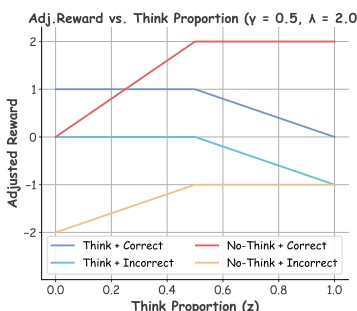
Figure 3: Effect of $z$ on $r_i^{\text{adj}}$.

While this reward structure encourages difficulty-aware behavior, it causes instability during early training. The model may collapse into a degenerate policy, either always thinking or always skipping, depending on which yields a higher expected reward in the short term. This limits exploration and hinders later optimization. To mitigate this, we introduce **batch-level reward balancing**:

Let $z \in [0, 1]$ denote the proportion of thinking trajectories **in a training batch**, and $1 - z$ the no-thinking proportion. A target balance ratio $\gamma \in (0, 1)$ and penalty slope $\lambda \geq 0$ control the strength of adjustment. For thinking and no-thinking samples, we compute soft penalty factors:

$$\delta_{\text{think}} = \min \left( 1, \max(0, (z - \gamma) \cdot \lambda) \right), \tag{3}$$
$$\delta_{\text{nothink}} = \min \left( 1, \max(0, (1 - z - \gamma) \cdot \lambda) \right). \tag{4}$$

Each sample $i$ is first assigned an original reward $r_i^{\text{naive}} \in \{+2, +1, 0, -1\}$ based on its thinking flag and correctness. The final adjusted reward is then:

$$r_i^{\text{adj}} = \begin{cases} (1 - \delta_{\text{think}}) \cdot r_i^{\text{naive}}, & \text{if think}_i = 1 \wedge \text{correct}_i = 1, \\ (1 - \delta_{\text{think}}) \cdot r_i^{\text{naive}} + \delta_{\text{think}} \cdot (-1), & \text{if think}_i = 1 \wedge \text{correct}_i = 0, \\ (1 - \delta_{\text{nothink}}) \cdot r_i^{\text{naive}}, & \text{if think}_i = 0 \wedge \text{correct}_i = 1, \\ (1 - \delta_{\text{nothink}}) \cdot r_i^{\text{naive}} + \delta_{\text{nothink}} \cdot (-2), & \text{if think}_i = 0 \wedge \text{correct}_i = 0. \end{cases} \tag{5}$$

The adjusted reward $r_i^{\text{adj}}$ introduces a soft, piecewise-linear modulation over the naive reward, resembling a hinge-like transformation. Figure 3 illustrates this behavior under a typical setting with $\gamma = 0.5$ and $\lambda = 2.0$. When thinking dominates ($z > \gamma$), the reward for thinking samples is softly reduced, especially for incorrect answers. Conversely, when no-thinking is overrepresented ($z \ll \gamma$), no-thinking rewards are suppressed. In both cases, the model is gently pushed to restore balance by favoring the less frequent behavior.

## 3.2 Stage 2: Reinforcing Reliable Behavior within Dual Modes

After establishing behavioral stability across thinking and no-thinking modes, the second stage focuses on improving task performance within each mode. Specifically, the objective is to enhance reasoning quality when invoked, and to promote accurate responses in the absence of thinking.

To allow the model to refine its behavior without external constraints, we remove the batch-level balancing used in the previous stage and allow free evolution of the reasoning policy. The reward is set directly to the naive definition:

$$r_i^{\text{adj}} = r_i^{\text{naive}}. \tag{6}$$

In this stage, we allocate a larger context budget during training, enabling longer responses when needed. Owing to the regularization established in Stage 1, the proportion of thinking in Stage 2 remains balanced, fluctuating naturally rather than collapsing.

### 3.3 Stage 3: Pruning Unnecessary Reasoning Paths via Length-Aware Reward

While the relaxed setup in Stage 2 improves accuracy, it also leads to overly long responses. Building on the stability established in prior stages, we now aim to improve reasoning efficiency.

Inspired by GRPO-LEAD[50], we introduce a length-aware reward modulation, encouraging brevity in no-thinking mode and rewarding elaboration only when warranted. Specifically, the adjusted reward in this stage is defined as:

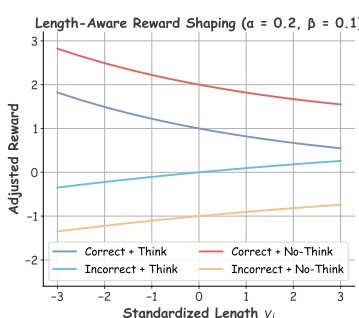

Length-Aware Reward Shaping (α = 0.2, β = 0.1)

$$r_i^{\text{adj}} = \begin{cases} r_i^{\text{naive}} + \left(-1 + e^{-\alpha y_i}\right), & \text{if correct}_i = 1, \\ r_i^{\text{naive}} + \left(1 - e^{-\beta y_i}\right), & \text{if correct}_i = 0. \end{cases} \tag{7}$$

where $y_i = \frac{L_i - \mu_q}{\sigma_q}$ is the standardized length of response $i$ within its query group $q$. Here, $L_i$ denotes the response length, while $\mu_q$ and $\sigma_q$ are the group-specific mean and standard deviation of lengths, computed separately for correct and incorrect sample groups. And $\alpha$ and $\beta$ are hyperparameters that control the sensitivity of the shaping term.

Figure 4: Effect of $\alpha, \beta$ on $r_i^{\text{adj}}$.

The reward decays with length for correct responses and grows for incorrect ones, encouraging concise success and thorough failure analysis, as an example illustrated in Figure 4. This final stage allows the model to adaptively regulate its reasoning depth, producing succinct responses without significantly compromising reliability.

## 4 Experiments

### 4.1 Setup

**Datasets and Models** We use the same training data as in DeepScaleR [17], comprising 40K mathematically problems with varying difficulties. Following prior works [49, 33], the evaluation is conducted on five standard math benchmarks: MATH, Minerva, Olympiad, AIME24, and AMC23. We evaluate the applicability of *AutoThink* on three R1-style models **with varying sizes and RL post-training status**: DeepSeek-R1-Distill-Qwen-1.5B/7B (abbreviated as Distill-R1-1.5B/7B), and DeepScaleR-Preview-1.5B [17] (abbreviated as DeepScaleR), the state-of-the-art 1.5B reasoning model obtained from Distill-R1-1.5B via context-extended RL at a training budget of up to $5,000.

**Baselines** We benchmark our approach against two classes of baselines designed to promote efficient reasoning. **(1) Prompt-only baselines**: we apply *standard* [9], *no-thinking* [19], and *ellipsis (ours)* prompting strategies on the base models, following the description illustrated in Figure 2a. **(2) RL-trained baselines**: including *Concise-RL* [6], *ShorterBetter* [44], and *ThinkPrune* [12], all of which aim to shorten reasoning traces by RL, but do not explicitly account for adaptive reasoning behavior. Among these methods, only *ThinkPrune* provides publicly available model checkpoints; we evaluate its two representative variants, `iter-2K` and `4K`. For *Concise-RL* and *ShorterBetter*, results are reported as published in their respective papers. **(3)** Additionally, we include a set of open-source

Table 1: (**Main Results**) Accuracy, Token Usage, and Efficiency Comparison Across Methods.

| Method | Accuracy (%) | | | | | | Token Usage | | | | | | E-F1(%) |
|---|---|---|---|---|---|---|---|---|---|---|---|---|---|
| | MATH | Minerva | Olympiad | AIME24 | AMC23 | AVG | MATH | Minerva | Olympiad | AIME24 | AMC23 | AVG | AVG |
| **Open-Source R1-Style Model: Train From DeepSeek-R1-Distill-Qwen-1.5B/7B or Even Larger** | | | | | | | | | | | | | |
| Open-RS3-1.5B | 83.0 | 26.3 | 43.3 | 30.6 | 63.0 | 49.2 | 5578 | 7579 | 11626 | 16651 | 11052 | 10497 | / |
| Still-3-1.5B | 84.9 | 28.4 | 45.0 | 31.0 | 64.6 | 50.8 | 4208 | 6021 | 9470 | 13399 | 8788 | 8377 | / |
| FastCuRL-1.5B | 87.9 | 30.9 | 49.8 | 40.8 | 72.3 | 56.7 | 3829 | 5849 | 7077 | 10300 | 6699 | 6571 | / |
| Light-R1-DS-7B | 92.1 | 37.6 | 58.1 | 62.3 | 82.4 | 66.5 | 3774 | 5434 | 8362 | 12064 | 7317 | 7390 | / |
| AReaL-boba-RL-7B | 93.4 | 37.7 | 62.1 | 65.4 | 85.7 | 68.9 | 4947 | 8290 | 10096 | 12905 | 8432 | 8934 | / |
| QwQ-32B | 95.1 | 45.3 | 69.0 | 76.7 | 95.5 | 76.3 | 5547 | 8650 | 9445 | 13970 | 4222 | 8367 | / |
| **Base Model: DeepSeek-R1-Distill-Qwen-1.5B** | | | | | | | | | | | | | |
| Standard Prompt | 83.1 | 26.0 | 43.7 | 27.5 | 62.5 | 48.6 | 5622 | 7688 | 11555 | 17322 | 10981 | 10633 | / |
| No-Thinking Prompt | 70.4 | 19.1 | 33.1 | 15.8 | 49.0 | 37.5 | 1256 | 628 | 2426 | 5793 | 2535 | 2528 | / |
| Ellipsis Prompt | 78.2 | 21.9 | 38.6 | 25.2 | 57.2 | 44.2 | 4194 | 4336 | 7752 | 13006 | 7980 | 7453 | / |
| Concise-RL | 81.0 | / | / | 30.0 | / | / | 1965 | / | / | 6752 | / | / | / |
| ShorterBetter | / | 27.6 | 38.4 | 20.0 | 56.6 | / | / | 1147 | 1814 | 2703 | 1946 | / | / |
| ThinkPrune-iter-2k | 82.6 | 28.1 | 43.6 | 26.7 | 64.9 | 49.2 | 1927 | 2126 | 3683 | 5806 | 3300 | 3368 | 9.9 |
| ThinkPrune-4k | 83.5 | 28.4 | 43.4 | 28.3 | 65.4 | 49.8 | 2723 | 3375 | 5504 | 8072 | 5040 | 4943 | 18.7 |
| *AutoThink-Stage1* | 79.4 | 21.4 | 40.5 | 27.7 | 59.0 | 45.6 | 3107 | 3867 | 7212 | 11673 | 6467 | 6465 | 0.0 |
| *AutoThink-Stage2* | 85.2 | 27.2 | 46.4 | 31.8 | 66.6 | 51.4 | 3702 | 5481 | 8030 | 12117 | 7415 | 7295 | 31.6 |
| *AutoThink-Stage3* | 84.0 | 28.1 | 44.8 | 34.6 | 67.0 | **51.7** | 2195 | 3212 | 5559 | 9514 | 5059 | 5108 | **39.6** |
| **Base Model: DeepScaleR-Preview-1.5B** | | | | | | | | | | | | | |
| Standard Prompt | 87.6 | 30.7 | 50.0 | 42.3 | 72.8 | 56.7 | 3171 | 4948 | 5967 | 9326 | 5675 | 5817 | / |
| No-Thinking Prompt | 78.1 | 21.8 | 40.9 | 23.8 | 58.4 | 44.6 | 1285 | 1217 | 2461 | 4682 | 2372 | 2403 | / |
| Ellipsis Prompt | 85.9 | 28.9 | 48.1 | 42.1 | 72.0 | 55.4 | 2890 | 4748 | 5416 | 9408 | 5095 | 5511 | / |
| ThinkPrune-iter-2k | 86.3 | 30.7 | 48.3 | 38.7 | 72.2 | 55.4 | 1838 | 2414 | 3254 | 5328 | 3166 | 3200 | 0.0 |
| ThinkPrune-4k | 86.5 | 30.6 | 48.5 | 36.5 | 71.8 | 54.8 | 2221 | 3039 | 4061 | 6624 | 3868 | 3963 | 0.0 |
| *AutoThink-Stage1* | 82.1 | 27.0 | 45.6 | 33.5 | 66.0 | 50.8 | 2473 | 5372 | 7328 | 12716 | 5440 | 6666 | 0.0 |
| *AutoThink-Stage2* | 87.6 | 31.8 | 50.1 | 42.9 | 73.9 | **57.3** | 2762 | 4315 | 5521 | 8567 | 5222 | 5277 | **7.5** |
| *AutoThink-Stage3* | 85.1 | 30.5 | 49.0 | 41.9 | 71.9 | 55.7 | 1897 | 3834 | 5005 | 9033 | 4696 | 4893 | 0.0 |
| **Base Model: DeepSeek-R1-Distill-Qwen-7B** | | | | | | | | | | | | | |
| Standard Prompt | 92.3 | 37.6 | 56.4 | 52.7 | 82.8 | 64.4 | 3928 | 5155 | 8815 | 13563 | 7613 | 7815 | / |
| No-Thinking Prompt | 78.2 | 22.1 | 40.2 | 22.7 | 53.7 | 43.4 | 722 | 486 | 1434 | 3269 | 1433 | 1496 | / |
| Ellipsis Prompt | 91.8 | 37.6 | 56.5 | 51.3 | 80.9 | 63.6 | 3752 | 4778 | 8643 | 13532 | 7616 | 7564 | / |
| Concise-RL | 90.3 | / | / | 51.7 | / | / | 2041 | / | / | 6632 | / | / | / |
| ShorterBetter | / | 44.1 | 50.7 | 53.3 | 75.9 | / | / | 1341 | 3410 | 5288 | 2580 | / | / |
| *AutoThink-Stage1* | 89.3 | 31.8 | 53.8 | 52.7 | 78.2 | 61.2 | 1763 | 1717 | 4798 | 8515 | 4397 | 4274 | 0.0 |
| *AutoThink-Stage2* | 92.2 | 38.5 | 56.2 | 57.1 | 83.7 | **65.5** | 2519 | 2980 | 5797 | 8676 | 4925 | 4979 | **7.2** |
| *AutoThink-Stage3* | 91.2 | 38.2 | 56.4 | 54.8 | 83.3 | 64.8 | 2146 | 2838 | 5498 | 8051 | 4645 | 4635 | 3.2 |

RL-finetuned models based on Distill-R1-1.5B/7B as reference, including *Open-RS3-1.5B* [5], *Still-3-1.5B* [22], *FastCuRL-1.5B* [28], *Light-R1-DS-7B* [38], *AReaL-boba-RL-7B* [21], and *QwQ-32B* [31]. **These models are not explicitly optimized for concise reasoning and differ significantly in both training objectives and computational budgets. We report their results for contextual reference only, aiming to highlight differences in design philosophy rather than to draw direct performance comparisons.**

**Training and Evaluation** All experiments are implemented using the `verl` framework [27], with most training hyperparameters retained at the default values. For all models, we set the batch size and training context length to $(128, 8K)$ in Stage 1, $(64, 16K)$ in Stage 2, and $(64, 24K)$ in Stage 3. We save model checkpoints at empirically selected steps based on observed convergence throughout the procedure: 220/440/130 for Distill-R1-1.5B, 110/240/60 for DeepScaleR, and 220/450/20 for Distill-R1-7B across Stages. During evaluation, all models use a 32K context window. We sample 16 rollouts per instance with temperature 0.6 and report the average pass@1 accuracy. Reward shaping hyperparameters are set to $\gamma = 0.5$, $\lambda = 2.0$ for Stage 1, and $\alpha = \beta = 0.05$ for Stage 3.

## 4.2 Main Results

Table 1 reports average accuracy and token usage across five mathematical benchmarks. To jointly evaluate reasoning accuracy and efficiency, we introduce the *Efficiency-F1 score* (E-F1), defined as:

$$\text{E-F1} = \left( 2 \cdot \frac{\Delta_{\text{acc}} \cdot \Delta_{\text{len}}}{\Delta_{\text{acc}} + \Delta_{\text{len}}} \right) \ \text{if acc} > \text{acc}_{\text{std}} \text{ and len} < \text{len}_{\text{std}}; \ \text{else } 0$$

where the normalized accuracy gain and token reduction are given by:

$$\Delta_{\text{acc}} = \frac{\text{acc} - \text{acc}_{\text{std}}}{\text{acc}_{\text{std}} - \text{acc}_{\text{no}}}, \quad \Delta_{\text{len}} = \frac{\text{len}_{\text{std}} - \text{len}}{\text{len}_{\text{std}} - \text{len}_{\text{no}}}$$

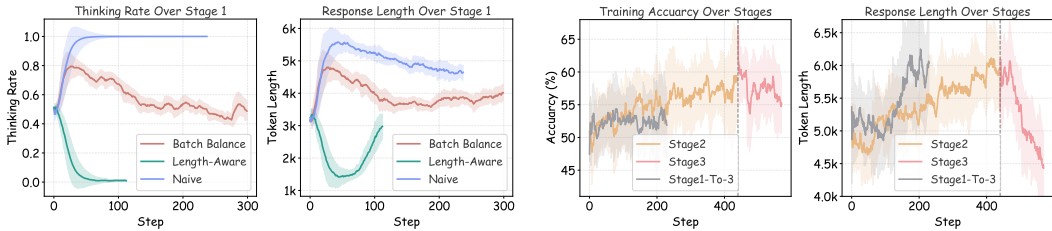

(a) Effect of Stage 1 reward design on reasoning.

(b) Effect of Stage 2 in performance boosting.

Figure 5: Prompting strategies shape reasoning behavior and computational cost.

The subscripts `std` and `no` refer to the *standard* and *no-thinking* baselines. A non-zero E-F1 indicates that the model improves upon the standard baseline in both accuracy and token usage, capturing the extent to which pruning enhances conciseness without degrading performance.

Despite the strong performance of existing open-source models, their outputs are substantially longer, **even reaching twice the length of ours at the same model size, suggesting that their gains stem from verbose reasoning but non-adaptive reasoning**. Prompt-based baselines (*no-thinking* and *ellipsis*) reduce length at the cost of accuracy. RL-based baselines also shorten outputs, but offer limited improvements on Distill-R1 and in some cases even reduce accuracy on DeepScaleR.

In contrast, *AutoThink* exhibits a staged progression in both accuracy and efficiency. **All three stages are consistently trained with the *ellipsis prompt* as the base prompting strategy.** Stage 1 primarily aims to stabilize the activation of reasoning behavior and has minimal impact on performance. Stage 2 leads to accuracy improvements over the standard prompt across all model backbones, demonstrating effective control over when to reason. Stage 3 introduces length-aware pruning, further reducing token usage while minimizing potential performance degradation. On Distill-R1-1.5B, *AutoThink-Stage3* achieves 51.7% accuracy with half the token usage of the *standard prompt* baseline. Remarkably, even on the heavily optimized DeepScaleR, *AutoThink-Stage2* further improves performance by 0.6 over the standard prompt while reducing token usage by an additional 10%. However, Stage 3 leads to a slight accuracy drop, likely because DeepScaleR has already undergone extensive optimization. This suggests that additional pruning may be unnecessary on fully optimized models.

### 4.3 Ablation Study

We conduct ablations on the reward design of our multi-stage RL framework on Distill-R1-1.5B to assess the necessity of each stage. The performance gains achieved by Stage 2 and the pruning effect of Stage 3 are already reflected in Table 1, in terms of accuracy and token usage. Here, we focus on two key aspects: (1) the role of batch reward balance in Stage 1, and (2) whether skipping Stage 2 and proceeding directly from Stage 1 to Stage 3 yields comparable performance.

**Batch Reward Balance Prevents Mode Collapse**  To assess the role of batch-level balancing in Stage 1, we examine its impact on stabilizing dual-mode reasoning behavior. Specifically, we plot the average thinking rate across training steps, as shown in Figure 5a. Under a naive reward, the model rapidly collapses into a thinking mode. Conversely, applying the length-aware reward (with $\alpha = 0.05$, $\beta = 0$) in naive reward to encourage brevity leads the model to collapse into a degenerate no-thinking mode. In contrast, the batch reward balance mechanism, by enforcing a target thinking ratio via penalty slope $\lambda$, helps stabilize training and supports the coexistence of thinking and no-thinking behaviors. **We observe that response length rises and then falls during training, indicating an increasing share of shorter, no-thinking responses.** These observations imply that the model implicitly performs reasoning pruning, akin to concise reasoning.

**Pruning Without Reinforcement Limits Performance**  We investigate the necessity of Stage 2 by applying Stage 3 directly after Stage 1, skipping the reinforcement phase. As shown in Figure 5b, the complete training pipeline that includes Stage 2 prior to Stage 3 yields a notable boost in both accuracy and response length, followed by effective pruning with minimal performance degradation. In contrast, bypassing Stage 2 results in stagnant accuracy and an eventual increase in response length after an initial decline. In contrast, skipping Stage 2 leads to stagnant accuracy and a rebound in response length. With comparable response lengths, the variant achieves only 47.6% accuracy across

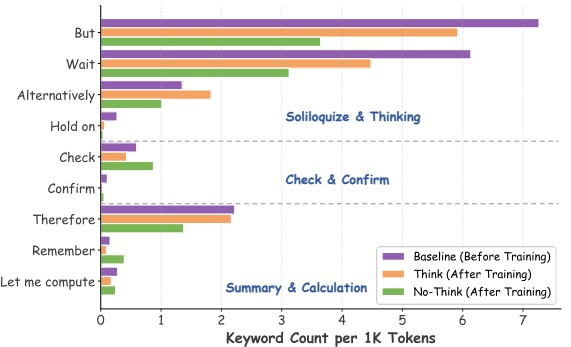
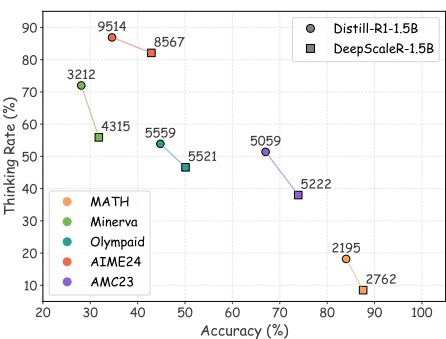

Figure 6: Keyword usage of reasoning behaviors across thinking and no-thinking modes.

Figure 7: Accuracy vs. Thinking Rate. The numbers indicate response lengths (tokens).

five benchmarks, notably lower than the 51.7% from full training. These observations underscore **the importance of Stage 2 in establishing stable and discriminative reasoning behaviors** that enable reliable pruning in the subsequent stage.

## 4.4 In-Depth Behavioral and Efficiency Analysis

**Lexical Patterns in Two Reasoning Modes** We analyze linguistic differences between thinking and no-thinking responses by quantifying the frequency of reasoning-related verbs (e.g., "Wait", "Alternatively", "Check") per 1,000 tokens, capturing how explicit reasoning is manifested in each mode. Following [12], we categorize these keywords into three functional groups on the MATH500 benchmark: (1) *Soliloquize & Thinking*, reflecting internal deliberation and self-correction, characteristic of R1-style reasoning; (2) *Check & Confirm*, indicating procedural verification; and (3) *Summary & Calculation*, marking final deduction and computational closure. As illustrated in Figure 6, *AutoThink* training substantially reduces soliloquy-like expressions, particularly under the no-thinking mode, indicating a decline in explicit internal deliberation. In contrast, verification and computation-related terms appear slightly more frequently in the no-thinking setting, suggesting a shift toward focused conclusion and validation rather than step-by-step verbalization.

**Correlation Between Task Difficulty and Reasoning Tendency** We investigate the relationship between the reasoning behavior and the inherent difficulty of the tasks. As shown in Figure 7, there exists a positive correlation between the thinking rate and task difficulty. To further quantify this relationship, we compute the average accuracy, thinking rate, and response length across all datasets. Here, accuracy serves as a proxy for dataset difficulty. The results indicate that, on more challenging datasets, models tend to invoke explicit reasoning more frequently and produce longer responses. This demonstrates that stronger models do not rely on explicit reasoning as frequently, yet outperform weaker models, highlighting an emergent ability to reason more selectively and efficiently.

**Readability and Accuracy of Dual Reasoning Modes** A common concern in reinforcement fine-tuning is that reward-driven optimization may degrade the fluency or coherence of generated reasoning traces. To assess whether *AutoThink* introduces such effects, we follow the evaluation setup in [12] and compute the perplexity (PPL) over the `<think>` span traces using Distill-R1-1.5B. For no-thinking variants, PPL is calculated over the segment following `</think>`. As shown in Table 2, the think mode of *AutoThink* maintains PPL comparable to standard prompting, while the no-think mode achieves the lowest PPL, reflecting more concise and fluent responses. Overall, all variants remain within acceptable readability ranges. Meanwhile, we analyze accuracy and token usage across reasoning modes. The results are also recorded in Table 2, **no-thinking responses are shorter and more accurate, suggesting effective handling of simpler problems. Thinking-mode responses are longer with slightly lower accuracy, reflecting allocation to harder cases.** These results indicate that *AutoThink* adaptively adjusts reasoning depth based on task difficulty.

**Evaluating *AutoThink* Under Standard and No-Thinking Prompts** We analyze how the trained model responds to the standard and forced-no-think prompts. The forced-no-think prompt is defined as `<think>\n...\n</think>\n\n`, which builds upon the *ellipsis* prompt but enforces an immediate termination of the thinking phase. The results of Distill-1.5B-*AutoThink* are presented in Table 3. As

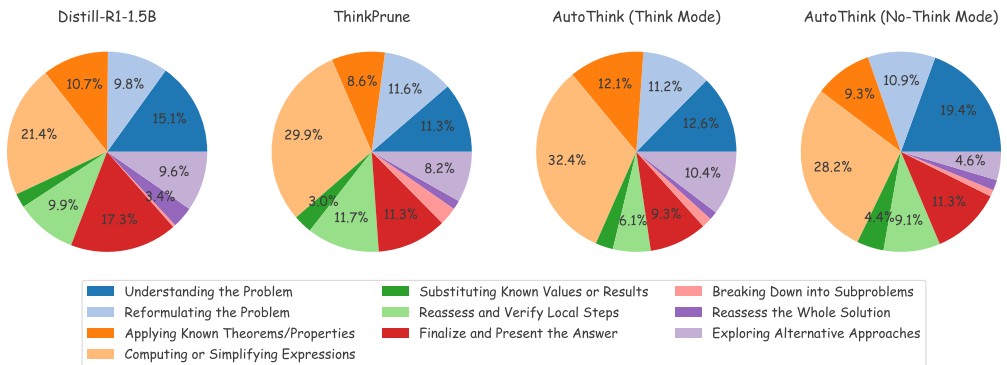

Figure 8: Distribution of Reasoning Behaviors Across Models and Reasoning Modes.

expected, the standard prompt induces longer reasoning traces and achieves higher accuracy, while the forced no-think prompt reduces token usage at the cost of slight performance degradation. These findings suggest that *AutoThink* has learned to internally compress its reasoning when appropriate, while retaining the ability to conditionally invoke reasoning via prompting.

Table 2: PPL, Acc & Token Length.

| Response | PPL | Acc (%) | Token |
|---|---|---|---|
| **Model: Distill-R1-1.5B** | | | |
| Standard Prompt | 1.61 | 83.1 | 5622 |
| No-Thinking Prompt | 1.87 | 70.4 | 1256 |
| AutoThink: Think Part | 2.29 | 56.4 | 5090 |
| AutoThink: No-Think Part | 1.50 | 90.1 | 1592 |
| **Model: DeepScaleR-1.5B** | | | |
| Standard Prompt | 2.19 | 83.1 | 5622 |
| No-Thinking Prompt | 1.85 | 70.4 | 1256 |
| AutoThink: Think Part | 2.43 | 63.9 | 5065 |
| AutoThink: No-Think Part | 1.84 | 89.2 | 1387 |

Table 3: *AutoThink* Performance on Three Prompts.

| Prompt | MATH | Minerva | Olympiad | AIME24 | AMC23 | AVG |
|---|---|---|---|---|---|---|
| **Accuracy (%)** | | | | | | |
| Ellipsis | 84.0 | 28.1 | 44.8 | 34.6 | 67.0 | 51.7 |
| Standard | 84.4 | 28.1 | 45.4 | 35.0 | 67.5 | 52.1 |
| Forced No-Think | 83.7 | 27.2 | 44.8 | 32.7 | 65.9 | 50.9 |
| **Token Usage** | | | | | | |
| Ellipsis | 2195 | 3212 | 5559 | 9514 | 5059 | 5108 |
| Standard | 2679 | 3534 | 5726 | 9862 | 5243 | 5408 |
| Forced No-Think | 2127 | 2877 | 5143 | 8668 | 4795 | 4722 |

**Reasoning Behavior Profiling** To gain a deeper understanding of how reasoning behaviors evolve, we annotate the generated solutions from each model with high-level problem-solving phases using GPT-4o. As illustrated in Figure 8, Distill-R1-1.5B distributes its reasoning effort across many surface-level activities, such as "reformulating the problem" and "understanding the problem." In contrast, ThinkPrune slightly shifts focus toward answer-finalization routines, while still exhibiting dispersed reasoning patterns. Notably, AutoThink in *Think Mode* allocates a larger proportion of steps to core reasoning phases, including "computing or simplifying expressions" and "applying known theorems," suggesting a more targeted and efficient reasoning trajectory. Meanwhile, in *No-Think Mode*, AutoThink maintains strong task comprehension and delivers concise outputs, dedicating most steps to problem understanding and direct computation. These findings indicate that AutoThink not only reduces redundant steps, but also adapts its reasoning structure based on the selected mode.

**Generality Beyond Mathematical Reasoning** To investigate whether AutoThink generalizes beyond mathematical reasoning, we additionally evaluate our models on three non-mathematical benchmarks: (i) **GPQA** for scientific multi-hop reasoning, (ii) **MMLU** for general multi-task language understanding, and (iii) **Live-Code-Bench** for code generation (20250727 release).

As shown in Table 4, AutoThink retains competitive accuracy while reducing token usage, indicating that adaptive reasoning behaviors extend beyond math tasks. Stage 2 even surpasses the baseline in accuracy while halving response length, highlighting the transferability of our approach to diverse domains.

Table 4: Performance of AutoThink on non-math benchmarks. Each cell shows Accuracy (%) / Avg. Length.

| | GPQA | MMLU | Live-Code-Bench | Avg |
|---|---|---|---|---|
| Distill-1.5B | 35.1 / 10026 | 49.5 / 2727 | 25.2 / 13372 | 36.6 / 8708 |
| AutoThink-Stage1 | 31.5 / 8889 | 47.7 / 1190 | 23.8 / 5653 | 34.3 / 5244 |
| AutoThink-Stage2 | 37.1 / 8617 | 48.8 / 1743 | 24.2 / 9647 | 36.7 / 6669 |
| AutoThink-Stage3 | 35.7 / 5659 | 48.8 / 1300 | 24.9 / 9054 | 36.5 / 5337 |

**Additional Analysis.** We further conduct additional analyses on more base models, hyperparameters, training cost, and case studies. Details are presented in Appendix B due to space limitations.

# 5 Related Works

**RL-based Post-Training for LLMs.** Reinforcement fine-tuning (RFT) has been widely adopted to improve the reasoning ability of LLMs [32, 13, 9, 11, 7, 34, 35]. Recent work on RL for LLMs has focused on improving the efficiency and effectiveness of large-scale RL training. Key techniques decoupling the clipping mechanism and introducing dynamic group sampling [46], mitigating value bias over long sequences [48, 47], difficulty-aware advantage reweighting [50], model ensembling [8] and designing minimal-form credit assignment strategies for rewards [4]. In addition, RFT has been shown to explicitly promote self-verification and self-correction behaviors [26, 18], while also supporting optimization of test-time compute [23]. Multi-stage, context-length-extended RL further amplifies the long-chain reasoning ability of R1-style models [17, 28]. In our work, RL is applied to train R1-style models to adaptively control their reasoning behavior, enabling selective thinking guided by multi-stage reward shaping.

**Mitigating Overthinking for LLMs.** While RFT improves performance, it may induce over-thinking, causing models to generate overly verbose reasoning with limited benefit [29, 14]. [3] address overthinking in R1-style models by using self-generated short CoT as positive signals in DPO, encouraging concise reasoning. [51] mitigate overthinking by training models to terminate with "I don't know" on unsolvable problems. Recent studies have shown that inserting pseudo-thinking cues into R1-style prompts [19], or manually controlling reasoning based on problem difficulty [10, 39, 15], can suppress the model's thinking behavior, but resulting in reduced performance. Other studies approach the problem from different perspectives: supervised fine-tuning (SFT) with short CoT responses [45, 20], incorporate response length–aware rewards in RFT [1, 44, 12, 6, 16], or leverage smaller models guide larger ones toward faster reasoning [15, 36]. Furthermore, [43] designs an early-exit mechanism during the reasoning phase to prevent excessive thinking along an incorrect chain of thought. Inspired by these findings, we first design a minimal prompt that elicits random thinking behavior, then apply multi-stage RL to guide the model to think adaptively based on task difficulty, without using external signals or teacher models.

# 6 Conclusion & Limitations

This work explores how R1-style LLMs can learn to reason adaptively. We propose *AutoThink*, a minimal prompting strategy paired with a multi-stage RL framework that enables task-aware thinking. Through stage-wise reward shaping, the model stabilizes reasoning patterns, reinforces effective behaviors, and prunes unnecessary steps. Experiments show that *AutoThink* achieves favorable accuracy–efficiency trade-offs, outperforming prompting and RL baselines without compromising performance, offering a scalable and controllable approach to efficient reasoning in LLMs.

While *AutoThink* demonstrates promising adaptive reasoning capabilities, several limitations remain:

**(1) Reward Hacking**: The model may bypass the separation between thinking and answering by embedding reasoning after the `</think>` tag. As shown in Figure 6, reasoning-related tokens still appear in no-thinking mode, suggesting incomplete behavioral separation.

**(2) Uncontrolled Reasoning Budget**: *AutoThink* adaptively decides when to think, but cannot control overall response length. Future work could explore budget-aware CoT generation, as seen in recent systems like Qwen3 [30].

**(3) Unfiltered Training Data**: We directly use the DeepScaleR dataset without filtering by task difficulty. Though simple data selection has shown utility, our focus lies in training design. Integrating curriculum-based filtering may further improve performance.

## Acknowledgments and Disclosure of Funding

This work is supported by the Strategic Priority Research Program of Chinese Academy of Sciences under Grant XDA0480302, Young Scientists Fund of The State Key Laboratory of Multimodal Artificial Intelligence Systems ES2P100112, National Natural Science Foundation of China 62402252, 62536003, and Guangdong High-Level Talent Programme 2024TQ08X283.

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

# Appendix

## A  Additional Definition and Prompts

### A.1  Definition of No-Thinking

In R1-style models (e.g., DeepSeek-R1), *Thinking* refers to generating explicit, step-by-step reasoning traces enclosed within `<think>` ⋯ `</think>`, enabling reflection and backtracking. By contrast, we define the *No-Thinking* mode [19] as immediately closing the `<think>` tag without producing any substantive reasoning, e.g., `<think> </think>` before moving to the final answer. This phenomenon, also referred to as *Non-Thinking* in the Qwen3 Technical Report [30], often emerges under our ellipsis prompt, which stochastically toggles the model between reasoning and shortcut modes and thus serves as a lightweight control signal for studying adaptive reasoning behaviors.

### A.2  Additional Prompt Variants

Beyond the prompt variants introduced in Section 2, we further explore an alternative strategy that explicitly encourages the model to self-select its reasoning behavior. Specifically, we augment the original CoT with **T**hink **B**y **D**ifficulty (*TBD*) prompt with an additional clause, *followed by ellipsis prompt* to preserve the optional-thinking behavior. As is shown below, where the *red* text highlights the added clause:

> *Let's think step by step and output the final answer within \boxed{}. Please decide whether to continue thinking based on the difficulty of the question.*

Despite appending the TBD prompt to explicitly encourage adaptive thinking, we observe no meaningful emergence of selective thinking behavior. As shown in Figure 9 and Table 5, we plot the no-thinking rate across difficulty levels (on MATH500) and report accuracy and token usage across five benchmarks. Interestingly, the addition of the TBD prompt leads to a slight drop in both accuracy and token consumption. This result suggests that prompting alone without any reinforcement signal is insufficient to reliably induce adaptive thinking behavior in Distill-R1 models.

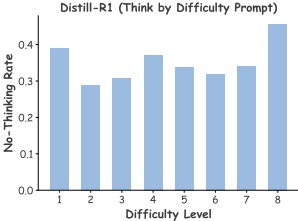

Figure 9: No-Thinking Rate.

| Prompt | MATH | Minerva | Olympiad | AIME24 | AMC23 | AVG |
|---|---|---|---|---|---|---|
| **Accuracy (%)** | | | | | | |
| Ellipsis Prompt | 78.2 | 21.9 | 38.6 | 25.2 | 57.2 | 44.2 |
| + TBD Prompt | 78.0 | 21.3 | 37.1 | 22.5 | 55.4 | 42.9 |
| **Token Usage** | | | | | | |
| Ellipsis Prompt | 4194 | 4336 | 7752 | 13006 | 7980 | 7453 |
| + TBD Prompt | 3893 | 3122 | 6490 | 11754 | 8796 | 6811 |

Table 5: Comparison Between Ellipsis and TBD Prompts.

## B  Extended Experimental Results

### B.1  Additional Results on Skywork-OR1-Math-7B

To further assess the generality of our method, we apply the AutoThink framework to Skywork-OR1-Math-7B[3], a state-of-the-art 7B model that achieves strong performance on mathematical reasoning tasks. Pretrained and fine-tuned with rule-based reinforcement learning on math and code tasks, this model represents one of the strongest 7B-scale math solvers. As shown in Table 6, the Ellipsis Prompt has limited effect on this highly optimized model, inducing only a marginal proportion of no-thinking responses, indicating reduced prompt sensitivity due to its deterministic reasoning policy.

Despite the limited prompt sensitivity, Stage 1 training with batch-level contrastive signals effectively captures and amplifies the model's latent no-thinking behavior, enabling more balanced reasoning

---

[3]https://huggingface.co/Skywork/Skywork-OR1-Math-7B

Table 6: Accuracy and Token Usage Comparison on Skywork-OR1-Math-7B.

| Method | Accuracy (%) | | | | | | Token Usage | | | | | |
|---|---|---|---|---|---|---|---|---|---|---|---|---|
| | MATH | Minerva | Olympiad | AIME24 | AMC23 | AVG | MATH | Minerva | Olympiad | AIME24 | AMC23 | AVG |
| **Base Model: Skywork-OR1-Math-7B** | | | | | | | | | | | | |
| Standard Prompt | 94.0 | 41.2 | 62.1 | 67.1 | 88.3 | 70.5 | 4669 | 7402 | 10102 | 14242 | 8849 | 9053 |
| No-Thinking Prompt | 85.5 | 26.1 | 48.1 | 45.6 | 68.2 | 54.7 | 1033 | 775 | 2982 | 6416 | 2402 | 2722 |
| Ellipsis Prompt | 94.0 | 41.8 | 61.8 | 69.1 | 88.0 | 70.9 | 4542 | 7399 | 10093 | 13813 | 8819 | 8933 |
| AutoThink-Stage1 | 92.9 | 36.9 | 59.3 | 65.4 | 86.6 | 68.2 | 1894 | 2177 | 4616 | 7068 | 4074 | 3966 |
| AutoThink-Stage2 | 94.0 | 40.0 | 62.3 | 63.1 | 88.9 | 69.7 | 2298 | 3247 | 5437 | 8091 | 4521 | 4719 |
| AutoThink-Stage3 | 93.1 | 38.8 | 61.1 | 62.7 | 88.2 | 68.8 | 1768 | 2287 | 4622 | 7372 | 3820 | 3974 |

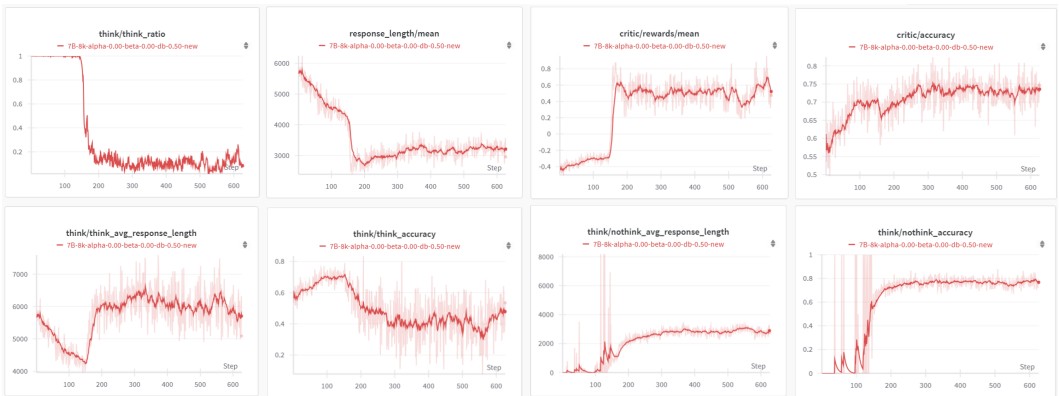

Figure 10: Training Curves of Skywork-OR1-Math-7B on Stage1.

patterns to emerge. Subsequent Stages 2 and 3 progressively refine this behavior. The full AutoThink framework is applied sequentially over three stages, trained for 600, 500, and 30 steps, respectively. **Notably, the final stage achieves a nearly 60% reduction in reasoning tokens (from 9053 to 3974), while preserving task accuracy with less than a 2% degradation compared to the standard prompting baseline.** These lightweight training phases are sufficient to induce substantial improvements in efficiency, even on strong pretrained models like Skywork-OR1-Math-7B.

We further visualize the training dynamics of Stage 1 in Figure 10, including the proportion of thinking responses, as well as the response length and accuracy stratified by thinking versus no-thinking behaviors. **At early stages, almost all responses involve explicit reasoning. However, batch-wise balancing gradually promotes the emergence of no-thinking behavior.** A clear modality shift occurs between steps 100 and 200, marked by a sharp increase in no-thinking responses, which directly contributes to the reduction in average response length. To explicitly encourage a balanced distribution between thinking and no-thinking responses throughout training, we set the target balance ratio $\gamma = 0.5$ in each of the three stages. Interestingly, while the accuracy of thinking responses slightly decreases during this phase, the overall accuracy continues to improve. This divergence suggests that **the model is learning to skip unnecessary reasoning on simpler problems**, thereby increasing both efficiency and decision quality through adaptive control over its reasoning mode.

## B.2  Additional Prompt Evaluation on Qwen3

We further extend our study to the **Qwen3-8B** [30] model by applying the proposed ellipsis prompting and adaptive training strategy. Table 7 reports accuracy and average token length across benchmarks. The results show that ellipsis prompting encourages a non-negligible amount of no-thinking behavior; however, this tendency does not align perfectly with task difficulty (e.g., AIME problems are significantly harder than MATH500, yet elicit a lower thinking rate). Together with observations on Skywork-OR1-Math-7B (where ellipsis prompting induced only ∼0.5% no-thinking behavior), these findings suggest that the AutoThink strategy can also induce autonomous reasoning behavior in Qwen3, with ∼13% occurrence of no-thinking responses under ellipsis prompting.

Table 7: Results of Qwen3-8B with different prompts. Each cell shows Accuracy (%) / Avg. Length.

| Qwen3-8B | MATH500 | Minerva | Olympiad | AIME24 | AMC23 | Avg |
|---|---|---|---|---|---|---|
| Standard Prompt | 97.0 / 5351 | 53.3 / 7010 | 73.5 / 11342 | 86.7 / 14690 | 88.1 / 10343 | 79.7 / 9747 |
| Ellipsis Prompt | 96.4 / 5109 | 49.5 / 5315 | 70.9 / 9891 | 68.3 / 13349 | 88.9 / 9858 | 74.8 / 8704 |
| No-Thinking Prompt | 84.1 / 1104 | 41.2 / 639 | 50.8 / 2860 | 26.3 / 6518 | 60.1 / 2913 | 52.5 / 2807 |
| Ellipsis Prompt: Thinking Rate | 96.9% | 67.5% | 89.0% | 87.5% | 96.2% | 87.4% |

## B.3  Hyperparameter Sensitivity

The three-stage framework is intentionally designed to be modular and interpretable, with each stage serving a distinct and simple role: (i) Stage 1 introduces a batch-wise reward balance to prevent mode collapse between thinking and no-thinking behaviors; (ii) Stage 2 focuses purely on reinforcing accuracy within each mode without additional reward shaping; (iii) Stage 3 adds length-aware shaping to encourage brevity for correct responses and elaboration for incorrect ones.

Among these stages, only Stage 1 and Stage 3 involve reward shaping beyond naive correctness. Even in these cases, the formulations remain straightforward and principled. Specifically, Stage 1 balances the modal ratio using a linear penalty controlled by hyperparameters $\gamma$ and $\lambda$, which are set to simple default values rather than finely tuned. As illustrated in Figure 3, the resulting reward curve naturally exhibits a symmetric form. Stage 3 reuses shaping terms $(\alpha, \beta)$ inspired by GRPO-LEAD, again without introducing any ad hoc modifications.

To further examine the robustness of these choices, we conduct sensitivity analyses of $\gamma$, $\lambda$, and $\alpha$ during training, with results summarized below.

**Stage 1 Parameters ($\gamma$, $\lambda$).**  In Stage 1, we balance the modal ratio between thinking and no-thinking behaviors using a linear penalty controlled by $\gamma$ and $\lambda$. These hyperparameters were not carefully tuned but set to commonly used values. To illustrate their effect, Table 8 reports the average thinking rate at steps 100 and 200 during training. Increasing $\gamma$ encourages more thinking trajectories, while larger $\lambda$ enforces stricter adherence to the target balance.

Table 8: Thinking rate (%) at checkpoints under different values of $\gamma$ and $\lambda$.

| | $\gamma$=0.5, $\lambda$=2 | $\gamma$=0.2, $\lambda$=2 | $\gamma$=0.8, $\lambda$=2 | $\gamma$=0.5, $\lambda$=1 | $\gamma$=0.5, $\lambda$=4 |
|---|---|---|---|---|---|
| Thinking-Rate@step100 | 62.4 | 57.8 | 99.9 | 71.4 | 51.7 |
| Thinking-Rate@step200 | 54.2 | 51.4 | 100.0 | 61.2 | 48.3 |

**Stage 3 Parameters ($\alpha$, $\beta$).**  In Stage 3, the shaping terms $(\alpha, \beta)$ control the rate of reward decay/growth with respect to response length. Table 9 shows response length at checkpoints under different $\alpha$ values with $\beta$ fixed at 0.05. Larger $\alpha$ accelerates length decay for correct responses, while smaller $\alpha$ relaxes the penalty.

Table 9: Response length under different $\alpha$ values ($\beta = 0.05$).

| | $\alpha = 0.05$ | $\alpha = 0$ | $\alpha = 0.1$ |
|---|---|---|---|
| Response-Length@step100 | 4734 | 6174 | 3623 |
| Response-Length@step200 | 4120 | 6322 | 2894 |

**Discussion.**  These results confirm that the shaping functions behave as intended: $\gamma/\lambda$ modulate the balance between modes in Stage 1, and $\alpha/\beta$ regulate the brevity of responses in Stage 3. Importantly, the overall training trends remain consistent with the main results, demonstrating the robustness of AutoThink without extensive hyperparameter tuning. Thus, while the overall pipeline appears multi-stage, each stage was deliberately designed with **minimal tuning and clear interpretability**. Looking ahead, it may be possible to **unify these stages through a more holistic reward formulation**, enabling the model to learn adaptive reasoning behavior within a single-stage process. We leave this as a promising direction for future work.

## B.4 Training Cost Comparison

We compare the training cost of *AutoThink* with two baseline methods, normalizing all runs to a unified batch size of 128. The results are shown in Table 10, *AutoThink* adopts a 3-stage schedule with increasing context lengths and a total of 500 steps, comparable to that of 540 in *ThinkPrune*.

In contrast, *ShorterBetter* trains in a single stage. While prior methods reduce context length to achieve compression, *AutoThink* expands it but prunes through shorter response length in Stage 3, resulting in comparable training cost. On H100 clusters with 4 nodes, training the all stages can be completed within one day for 1.5B models, and 2.5 days for 7B.

Table 10: Training Cost of Distill-R1-1.5B.

| Method | Steps (Batch Size=128) | Context Length |
|---|---|---|
| AutoThink | $\approx 220 + 220 + 60 = 500$ | 8K / 16K / 24K |
| ThinkPrune | $\approx 80 + 180 + 180 = 540$ | 4K / 3K / 2K |
| ShorterBetter | $\approx 300$ | 6K |

Table 11 provide estimated GPU-hour costs for all methods using Distill-R1-1.5B as the base model, *AutoThink* operates within the same order of compute as concise baselines such as ThinkPrune and ShorterBetter, yet achieves notably stronger performance. In contrast, DeepScaleR, which primarily aims to maximize performance, requires more than $3\times$ higher compute due to its longer context length and increased RL iterations.

Table 11: Estimated GPU-hour cost of training Distill-R1-1.5B with different methods on H100.

| Method | GPU Hours | Avg ACC / Length |
|---|---|---|
| ThinkPrune-iter-2K | ~400 | 49.2 / 3368 |
| ShorterBetter | ~200 | 44.7 / 1915 |
| **Distill-1.5B-AutoThink** | ~700 | 51.7 / 5108 |
| DeepScaleR | ~2200 | 56.7 / 5817 |
| **DeepScaleR-1.5B-AutoThink** | ~250 | 57.3 / 5277 |

## C  Addressing Potential Challenges

While AutoThink demonstrates robust improvements, several open challenges remain in the area of reasoning control. We outline possible directions for addressing these limitations below:

**Token-Budget Control.**  Token-budget constraints have been partially explored in prior works [1, 44], where budget-aware reward functions penalize excessively long completions. Such formulations can be readily integrated with AutoThink in a plug-and-play manner to enforce global compute budgets and further improve efficiency.

**Dataset Noise.**  The presence of noise in large-scale reasoning datasets can hinder training efficiency. Prior studies [28] suggest that curriculum learning or filtering samples by correctness or difficulty can improve learning quality. These strategies are orthogonal to our reward design and could be combined with AutoThink to further enhance robustness.

**Reward Hacking.**  A common issue is reward hacking, where the model continues reasoning after the `</think>` tag. This can be mitigated by explicitly penalizing reasoning-related patterns outside the `<think>` span, or by rewarding clean separation between thought and final answer. Both strategies can be incorporated into future iterations of our reward function.

Overall, we view these solutions as complementary and composable with our framework. Future work will explore tighter integration of these mechanisms to provide a more comprehensive solution to reasoning control.

## D  Case Study

Figure 11,12,13 presents some examples comparing four prompting strategies. For easy problems, *AutoThink* produces correct answers without explicit reasoning, reflecting effective fast thinking. For medium problems, it may activate both reasoning modes, with thinking and no-thinking responses potentially coexisting. For hard problems, the model engages in deeper, slower reasoning, demonstrating iterative understanding and self-verification before arriving at the correct solution. These observations demonstrate how *AutoThink* adapts its reasoning to problem difficulty, balancing efficiency and reliability through dynamic control of reasoning depth.

**Question**: What is the result when the greatest common factor of 6432 and 132 is increased by 11?

**Standard Prompt, 2125 tokens**

**[Understand the Problem, 229 tokens]** Okay, so I have this math problem here: ... by using the Euclidean algorithm ... The last non-zero remainder is the GCF.

**[Attempt to Solve, 456 tokens]** Alright, so let's apply that to 6432 and 132. First ... 6432 mod 132 ... 132 mod 96 ... 96 mod 36 ... 36 mod 24 ... Therefore, the GCF of 6432 and 132 is 12.

**[Reassess the Whole Solution, 1146 tokens]** Wait, hold on a second ... GCF = 12 ... Therefore, the result is 23. Wait, hold on ... gives 23. But just to be absolutely thorough ... So, the final answer is 23.

**[Finalize and Present the Answer, 294 tokens]** To find the result ... Therefore, the result is 23.

**No-Thinking Prompt, 196 tokens**

**[Solve the Problem by CoT, 184 tokens]** To find ...
1. Find the GCF of 6432 and 132 ... GCF=...=12
2. Increase the GCF by 11: ... 12+11=23.

**[Present the Answer, 12 tokens]** Final Answer: 23.

**AutoThink: No-Thinking Mode, 455 tokens**

**[Understand the Problem, 97 tokens]** Okay, so I need to find ... I can use the Euclidean algorithm ...

**[Solve the Problem by CoT, 345 tokens]** Starting with 6432 and 132 ... 6432 mod 132 ... 132 mod 96 ... 96 mod 36 ... 36 mod 24 ... So, the GCF of 6432 and 132 is 12 ... 12 + 11 = 23.

**[Present the Answer, 13 tokens]** The final result is 23.

Figure 11: Easy Case: *AutoThink* solves the problem via no-thinking mode with few tokens.

**Question**: Let $z = 2 + \sqrt{2} - \left(3 + 3\sqrt{2}\right)i$, and let $c = 2 - 3i$. Let $w$ be the result when $z$ is rotated around $c$ by $\pi/4$ counter-clockwise. Find $w$.

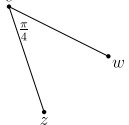

**Standard Prompt, 3897 tokens**

**[Understand the Problem, 186 tokens]** Alright, so I have this problem ... Let me try to recall the exact steps.

**[Attempt to Solve, 1200 tokens]** First, I think I need to translate $z$ so that $c$ is at the origin ... Okay, next step is to rotate ... Now, combine all these ... Thus, $w = 6 - 5i$.

**[Reassess the Whole Solution, 1969 tokens]** Wait, hold on ... $6 - 5i$. But wait ... $6 - 5i$. But just to make sure ... $6 - 5i$. So, yes ... $6 - 5i$. But just to make sure ... $6 - 5i$. Therefore, all my computations seem consistent.

**[Finalize and Present the Answer, 542 tokens]** Given $c = 2 - 3i$ ... Thus, the final result is: $6 - 5i$.

**No-Thinking Prompt, 308 tokens**

**[Solve the Problem by CoT, 289 tokens]** To find ...
1. Translate the complex plane ... $z' = \cdots = \sqrt{2} + 0i$.
2. Rotate $z$ by $\pi/4$ ... $w' = \cdots = 1 + i$.
3. Translate back ... $w = \cdots = 3 - 2i$.

**[Present the Answer, 19 tokens]** Final Answer: $3 - 2i$.

**AutoThink: Thinking Mode, 2841 tokens**

**[Understand the Problem, 284 tokens]** Okay, so I need to ... So, the steps are: 1. 2. 3. ... Let me write that out in equations.

**[Attempt to Solve, 793 tokens]** First, ... Now, I need to rotate ... Wait, hold on ... Now, we need to translate back ... So, $w = -2i$.

**[Reassess the Whole Solution, 1275 tokens]** Wait, that seems straightforward ... So, $z'' = 4 - 2i$. Wait, that contradicts ... $w = 6 - 5i$. Wait, that's different from ... Wait, so the final answer is $6 - 5i$ ... So, to recap: ... Therefore, the final answer is $6 - 5i$.

**[Finalize and Present the Answer, 489 tokens]** Given $c = 2 - 3i$ ... Thus, the complex number $w$ is: $6 - 5i$.

**AutoThink: No-Thinking Mode, 1013 tokens**

**[Understand the Problem, 145 tokens]** Okay, so I need to find the point $w$ ... First, let's write down the given points: ...

**[Solve the Problem by CoT, 836 tokens]** To rotate ... I can use the rotation formula ... So, $e^{i\theta} = \cdots = \frac{\sqrt{2}}{2} + i\frac{\sqrt{2}}{2}$. Let me compute $z - c$ first ... Now, multiply this by ... Let me compute this multiplication: ... Now, add back ... $w = \cdots = 6 - 5i$.

**[Present the Answer, 32 tokens]** Therefore, the result $w$ is $6 - 5i$.

Figure 12: Medium Case: *AutoThink* exhibits both thinking and no-thinking modes on the problem.

**Question**: In quadrilateral ABCD, angle BAD and angle CDA are trisected as shown. What is the degree measure of angle AFD?

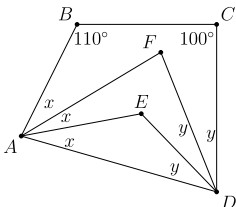

**No-Thinking Prompt, 360 tokens**

[Solve the Problem by CoT, 344 tokens] To determine the measure of angle AFD ...
1. Identify the Triangles: ...
2. Sum of Angles: ... so angle ABD is 140°
3. Determine Angle AFD : ... However, this result exceeds the possible measure ... Reassessing, ... Totaling 20°

[Present the Answer, 16 tokens] Final Answer: 20°.

**Standard Prompt, 13357 tokens**

[Understand the Problem, 1575 tokens] Alright, so I have this geometry problem here with quadrilateral ABCD ... So, it's a trapezoid? Wait ... Or maybe ... From the Asymptote code ... Wait, BC is ... Wait, the Asymptote code ... Wait, but angles at B and C ... But I don't know if that's directly helpful.

[Attempt to Solve, 662 tokens] Wait, but we need to find angle AFD ... Wait, actually, ... But maybe we can use the Law of Sines or Law of Cosines ... we need to know the coordinates or the lengths.

[Explore Other Approaches, 10672 tokens] So, let's compute angle BAD first ... Wait ... Wait ... maybe we can use Ceva's Theorem ... Wait, maybe better to consider triangle AFD ... Wait, angle FAD is angle between AF and AD ... but I don't know angle FDB. ... something ... Wait, in triangle ABD ... Hmm, not sure ... perhaps I can instead consider that angle AFD is 30°. But let me think again ... Wait, perhaps using vectors ... Wait, but since angle BAD is 3x ... Wait, not sure. ... Wait ... But how does this help us? ... so I will go with 30°...

[Finalize and Present the Answer, 448 tokens] In quadrilateral ABCD ... Thus, the degree measure of angle AFD is 30.

**AutoThink: Thinking Mode, 7718 tokens**

[Understand the Problem, 589 tokens] Okay, so I have this geometry problem here about quadrilateral ABCD... So, AFD is the angle at point F between points A and D.

[Attempt to Solve, 1515 tokens] To find angle AFD, maybe I can ... Wait, but the Asymptote code is just for drawing ... So, angle BAD is 110 degrees ... Wait, maybe I'm complicating it ... Given that, let me calculate the coordinates ... So, theta ≈ 80.8 degrees.

[Explore Other Approaches, 5072 tokens] Hmm, so angle AFD is approximately 81 degrees ... But let me check my calculations again ... Alternatively, maybe I made a mistake ... Let me compute it more accurately ... maybe it's 80 degrees ... Wait, maybe I can consider the entire quadrilateral and the angles ... But wait, in the Asymptote code ... So, perhaps angle AFD is 80 degrees or 81 degrees. Wait, ... I think it's 81 degrees ... I think 80 degrees is the answer ... So, the answer is 80.

[Present the Answer, 542 tokens] To solve the problem ... the degree measure of angle AFD is 80.

Figure 13: Hard Case: *AutoThink* solves the problem via thinking mode with repeated verification.

