# OpenReview forum: "Learning When to Think: Shaping Adaptive Reasoning in R1-Style Models via Multi-Stage RL"
_NeurIPS.cc/2025/Conference — NeurIPS 2025 poster_

### Official Review · Reviewer_L1QB · 2025-06-09

**Clarity:** 4
**Significance:** 3
**Originality:** 4
**Rating:** 5
**Confidence:** 4

**Summary:**

The paper introduces AutoThink, a novel framework designed to imbue large reasoning models (LRMs) with adaptive thinking capabilities. Addressing the common problem of "over-thinking" in CoT-style models, which leads to computational overhead, AutoThink builds upon a unique observation: a simple ellipsis ("...") inserted into the prompt of R1-style models can stochastically trigger either a detailed "thinking" mode or a concise "no-thinking" mode. Leveraging this latent controllability, the authors propose a three-stage reinforcement learning (RL) framework to progressively optimize the model's reasoning policies. Specifically, Stage 1 stabilizes the coexistence of these dual modes, Stage 2 reinforces accurate reasoning within each mode, and Stage 3 prunes redundant reasoning via length-aware rewards. Experimental evaluations on five mathematical benchmarks demonstrate that AutoThink achieves superior accuracy-efficiency trade-offs compared to existing prompting and RL-based pruning methods, notably improving accuracy while significantly reducing token usage, and is seamlessly integrable into various R1-style models.

**Questions:**

1. I would like to know the relationship between **RL steps and acc** under three different reward functions (Fig. 5), which, when combined with the existing analysis of **RL steps and tokens** (Fig. 5a), could help readers better understand the comprehensive impact of these different reward functions.

2. Why did the training of the `Length-Aware reward function` stop at around step 110 in Fig. 5a? Specifically, I noticed that the token usage increases after approximately step 50. What will happen when its training were continued?

**Ethical Concerns:**

["NO or VERY MINOR ethics concerns only"]

**Final Justification:**

The detailed experiment in the rebuttal has addressed my main concerns.

**Limitations:**

Yes

**Paper Formatting Concerns:**

No formatting issues in this paper.

**Quality:**

3

**Strengths And Weaknesses:**

# Strengths
1. The finding that a minimalistic, non-semantic token like an ellipsis can stochastically activate distinct reasoning modes in R1-style models is highly original and provides a surprisingly simple yet effective control mechanism.
2. The three-stage RL design is well-conceived and robust. Each stage systematically addresses specific challenges in developing adaptive reasoning, and the ablation studies effectively validate the necessity and contribution of each phase.
3.  AutoThink consistently demonstrates favorable accuracy-efficiency trade-offs across a diverse set of mathematical benchmarks and different R1-style models. The substantial reduction in token usage (e.g., 52% on Distill-R1-1.5B) without sacrificing accuracy is a compelling practical advantage. The introduction of the E-F1 metric is also a valuable contribution for evaluating this trade-off.

# Weaknesses
1. While impressive, the specific behavior triggered by the ellipsis prompt might be highly specific to the architectural nuances and pre-training of R1-style models. It is unclear how transferable this phenomenon is to other LLM architectures or different Chain-of-Thought formats, which might limit the generality of this specific control handle. Further theoretical or empirical investigation into why the ellipsis works could strengthen this aspect.
2.  This paper introduces several key hyper-parameters, such as $\gamma$, $\lambda$ in Stage 1, $\alpha$, $\beta$ in Stage 3. However, the authors only state the empirical values without providing any sensitivity analysis for different setting of these parameters. This makes it difficult for readers to understand the impact of different settings.

---

> ### Author Rebuttal · Authors · 2025-07-30
>
> We sincerely thank you for the encouraging and constructive feedback. We are glad that you found the motivation, prompt insight, and stage-wise design of AutoThink to be elegant and effective. We address your concerns below:
>
> ---
>
> **1. On the Generality of the Ellipsis Prompt and Applicability to Other Models**
>
> Our method is specifically designed for Large Reasoning Models (R1-style models),
>
> Thank you for this valuable comment. We would like to clarify that our method specifically targets at **Large Reasoning Models**, where *Thinking* refers to generating a step-by-step reasoning trace within the `<think> {thinking_content} </think>` tag, typically involving continuous reflection and backtracking. This structured reasoning format has been adopted in recent large reasoning models with explicit `<think>`-style reasoning blocks to modularize thought and improve interpretability.
>
> Among open-source models, **DeepSeek-R1-Distill-Qwen** is a representative R1-style distilled model trained from Qwen2.5 by using DeepSeek-R1-671B reasoning data, making it a natural choice for our main experiments.
>
> To further validate generality, we have already included **DeepScaleR** in our experiments (Table 1), which is built upon DeepSeek-R1-Distill-Qwen-1.5B and strengthened via multi-stage context extension and RL. Even though DeepScaleR has undergone extensive fine-tuning to favor long reasoning chains, we observe that under the **ellipsis prompt**, the average token usage still drops from 5817 to 5511, suggesting that our prompt design still elicits a small amount of no-thinking behavior. Moreover, **AutoThink stages applied on top of DeepScaleR** demonstrate clear ability to further reduce token usage with minimal or no accuracy loss (see Table 1).
>
> In addition, we conducted further validation on **Skywork-OR1-Math-7B**, a stronger RL-trained R1-style model that has undergone deeper optimization for mathematical and code reasoning. We report below the results on different prompting/training stages:
>
> |Skywork-OR1-Math-7B|Accuracy (%)|Token Usage|
> |-|-|-|
> |Standard prompt|70.5|9053|
> |Ellipsis Prompt|70.9|8933|
> |No-Thinking Prompt|54.7|2722|
> |AutoThink-Stage1|68.2|3966|
> |AutoThink-Stage2|69.7|4719|
> |AutoThink-Stage3|68.8|3974|
>
> Although **ellipsis prompting alone induces minimal no-thinking behavior (only ~0.5% occurrence), our method successfully encourages such behavior to emerge through structured reward shaping**, reducing the average token usage from 8933 to 3966 in Stage 1 while maintaining comparable accuracy. During Stage-1 training, we observed a sharp drop in thinking ratio and response length around step 120, indicating a successful emergence of dual-mode balance.
>
> Finally, due to time and compute constraints during the rebuttal phase, we were unable to run the full training pipeline on **Qwen3**. However, we conducted a preliminary evaluation to assess the effectiveness of ellipsis prompting, recording both accuracy and token length:
>
> |Qwen3-8B|MATH500|Minerva|Olympaid|AIME24|AMC23|Avg|
> |-|-|-|-|-|-|-|
> |Standard prompt|97.0/5351|53.3/7010|73.5/11342|86.7/14690|88.1/10343|79.7/9747|
> |Ellipsis Prompt|96.4/5109|49.5/5315|70.9/9891|68.3/13349|88.9/9858|74.8/8704|
> |No-Thinking Prompt|84.1/1104|41.2/639|50.8/2860|26.3/6518|60.1/2913|52.5/2807|
> |Ellipsis Prompt: Thinking Rate|96.9%|67.5%|89.0%|87.5%|96.2%|87.4%|
>
> The use of the ellipsis prompt can encourage a certain degree of no-thinking behavior, and this behavior does not correlate with task difficulty (AIME problems are significantly harder than those in MATH500, yet elicit a lower thinking rate). **Based on the experimental results from the above Skywork-OR1-Math-7B (about 0.5% occurrence of no-thinking behavior with Ellipsis Prompt), we believe that the AutoThink strategy could also induce autonomous reasoning in Qwen3 (~13% occurrence of no-thinking behavior with Ellipsis Prompt).**
>
> ---
>
> **2. Hyperparameter Sensitivity**
>
> We appreciate your concern regarding the complexity of our method. We would like to clarify that the **three-stage structure is intentionally modular and interpretable**, with each component driven by distinct and simple design principles:
>
> * **Stage 1** introduces a batch-wise reward balance to prevent mode collapse between thinking and no-thinking behaviors.
> * **Stage 2** focuses purely on reinforcing accuracy within each mode (no additional reward shaping).
> * **Stage 3** adds length-aware shaping to encourage brevity for correct responses and elaboration for incorrect ones.
>
> Among the three stages, only Stage 1 and Stage 3 involve reward design beyond naive correctness. Even then, the reward formulations are simple and intuitive:
>
> * In Stage 1, we balance the modal ratio using a linear penalty with hyperparameters γ and λ. **These parameters were not carefully tuned, but simply set to commonly used values.** As shown in Figure 3, the reward curve exhibits a natural symmetric form.
> * In Stage 3, we reuse default shaping terms (α, β) **inspired by GRPO-LEAD, additional any revision**.
>
> We have already provided **ablations in Section 4.3 and Figure 5**, which show:
>
> * Without Stage 1, the model collapses into a single mode (thinking or no-thinking).
> * Skipping Stage 2 results in a clear drop in final accuracy (51.7% → 47.6%).
>
> **We acknowledge the lack of hyperparameter sweeps in the main paper. In additional experiments during the rebuttal phase, we find**:
>
> * Increasing **γ** encourages more thinking behavior in Stage 1.
> * Higher **λ** enforces stricter adherence to the target balance.
> * Larger **α/β** in Stage 3 accelerate reward decay/growth in response_length.
>
> To illustrate these trends, we record the *thinking ratio* measured as the running average over the most recent 20 steps at two key checkpoints (step 100 and step 200) during Stage 1 training:
>
> ||γ=0.5, λ=2|γ=0.2, λ=2|γ=0.8, λ=2|γ=0.5, λ=1|γ=0.5, λ=4|
> |-|-|-|-|-|-|
> |Thinking-Rate@step100|62.4|57.8|99.9|71.4|51.7|
> |Thinking-Rate@step200|54.2|51.4|100.0|61.2|48.3|
>
> Additionally, we record the response lengths in Stage 3 under different values of α, with β fixed at 0.05:
>
> ||α=0.05|α=0|α=0.1|
> |-|-|-|-|
> |Response-Length@step100|4734|6174|3623|
> |Response-Length@step200|4120|6322|2894|
>
> These training trends are consistent with the intended behavior of the shaping functions. We will include these findings in the appendix and release them alongside the camera-ready version.
>
> ---
>
> **3. On the Reward Impact and Early Stopping in Figure 5a**
>
> We appreciate your question regarding the relationship between RL steps and accuracy in Stage 1. First, we would like to clarify that the primary objective of Stage 1 is to stabilize the coexistence of *thinking* and *no-thinking* modes, rather than to directly improve accuracy. The effects of different reward coefficients have already been discussed in detail in the previous section.
>
> In Figure 5a, we observe that all three reward variants generally show upward trends. Among them, the **naive reward** increases slightly faster than the **batch balance reward**, as it implicitly encourages fully "thinking" responses to achieve higher accuracy.
>
> In contrast, the **length-aware reward** exhibits a non-monotonic pattern: it decreases initially and then gradually increases. This trend aligns closely with the change in response length. Our interpretation is that the length-aware reward encourages shorter, more concise answers. In the early stages of training, **when the model’s overall accuracy is still low, responses with short and no-thinking mode tend to receive higher rewards**, leading the model to collapse into a no-thinking mode. As training progresses, however, **longer no-thinking responses tend to be more accurate, guiding the policy to optimize toward generating longer responses**, resulting in a moderate increase in response length.
>
> We observed that after step 110, training under the length-aware reward leads to a **stable response length range**. However, since the model remains largely no-thinking single-modal, its accuracy is generally lower than that of models trained with the batch-balance or naive reward.
>
> To improve clarity, we will include RL step–accuracy curves and more detailed discussions in the camera-ready version, offering a clearer picture of how each reward function influences learning dynamics.
>
> ---
>
> We sincerely hope that these clarifications and additional results help address your concerns. If you have any additional feedback or suggestions, we would greatly welcome them and be happy to address them in detail.

---

> ### Comment · Reviewer_L1QB · 2025-08-04
>
> Thanks for your feedback. You have clarified my concerns. I will raise my score.

---

> > ### Author Response · Authors · 2025-08-04
> >
> > We are grateful for your valuable feedback and the time you took to engage with our work

---

### Official Review · Reviewer_x9j5 · 2025-07-01

**Clarity:** 4
**Significance:** 3
**Originality:** 3
**Rating:** 5
**Confidence:** 4

**Summary:**

This paper addresses the over-thinking problem in R1-style large reasoning models (LRMs), where explicit step-by-step reasoning incurs high computational costs for simple tasks. The authors introduce AutoThink, a novel framework that leverages an ellipsis prompt to trigger stochastic thinking/no-thinking modes in LRMs, combined with a three-stage reinforcement learning (RL) framework for adaptive reasoning. The three stages include:
- Stage 1: Stabilizing dual-mode coexistence to prevent policy collapse.
- Stage 2: Reinforcing accurate reasoning within both modes.
- Stage 3: Pruning redundant steps via length-aware rewards.

AutoThink demonstrates superior accuracy-efficiency trade-offs on five mathematical benchmarks compared to prompting and RL-based pruning methods. The study also validates the effectiveness of difficulty-aware reasoning and the necessity of each training stage through ablation studies.

**Questions:**

While validated on mathematical benchmarks, does AutoThink extend to non-mathematical reasoning tasks?

**Ethical Concerns:**

["NO or VERY MINOR ethics concerns only"]

**Final Justification:**

The authors’ rebuttal provided additional non-mathematical benchmark results (GPQA, MMLU, Live-Code-Bench), which, while preliminary, indicate that AutoThink’s adaptive reasoning control generalizes beyond the math domain. This directly addresses my main question about broader applicability. The explanation regarding the multi-stage RL pipeline clarified that the design is purposeful, minimally tuned, and interpretable, which alleviates concerns about unnecessary complexity. The paper remains technically solid, presents a well-motivated and validated framework, and shows strong empirical gains in accuracy-efficiency trade-offs. Given its relevance, solid evaluation, and potential for broad impact, I maintain the accept rating of 5.

**Limitations:**

Please refer to the weakness section.

**Quality:**

3

**Strengths And Weaknesses:**

### Strengths
- The progressive reward shaping in AutoThink aligns with intuitive reasoning principles (e.g., balancing thinking modes, reinforcing correctness, pruning redundancy). Extensive experiments and ablation studies (e.g., mode collapse prevention, stage dependency) thoroughly validate the framework’s efficacy.
- The research tackles a critical limitation of modern LRMs—excessive computational overhead from unnecessary reasoning—making it highly relevant for practical deployment in resource-constrained scenarios.
### Weaknesses
- The multi-stage RL pipeline and tailored reward functions are relatively complex.

---

> ### Author Rebuttal · Authors · 2025-07-30
>
> We sincerely thank you for the positive and thoughtful evaluation of our work. We're especially grateful for your recognition of AutoThink’s contribution to improving reasoning efficiency and your appreciation of the progressive reward shaping design. We address your concerns below:
>
> ---
>
> **1. Generality Beyond Mathematical Reasoning**
>
> We appreciate the your concern regarding generalization beyond math tasks. In response, we have conducted additional evaluations of our AutoThink models on three non-mathematical benchmarks:
>
> * **GPQA** for scientific multi-hop reasoning,
> * **MMLU** for general multi-task language understanding,
> * **Live-Code-Bench** for code generation, we use the newest 20250727 version.
>
> The results show that AutoThink-trained models **retain competitive accuracy** while reducing average token usage, suggesting that the learned **adaptive reasoning behaviors generalize beyond math**.
>
> ||GPQA|MMLU|LiveCodeBench|Avg|
> |-|-|-|-|-|
> |DeepSeek-R1-Distill-1.5B|35.1/10026|49.5/2727|25.2/13372|36.6/8708|
> |AutoThink-Stage1|31.5/8889|47.7/1190|23.8/5653|34.3/5244|
> |AutoThink-Stage2|37.1/8617|48.8/1743|24.2/9647|36.7/6669|
> |AutoThink-Stage3|35.7/5659|48.8/1300|24.9/9054|36.5/5337|
>
> Although these findings are preliminary, they suggest that math-domain training can endow LLMs with transferable control over reasoning depth. In future work, we will explore domain-specific training to further strengthen AutoThink’s adaptability in broader tasks.
>
> ---
>
> **2. Complexity of the multi-stage RL**
>
> Regarding your comment on the **complexity of the multi-stage RL pipeline and tailored reward functions**, we would like to emphasize that each component was introduced to serve a clear, intuitive purpose—as you summarized precisely:
>
> * **Stage 1** ensures balanced coexistence between thinking and no-thinking modes via a simple linear penalty mechanism. The associated hyperparameters (γ, λ) were **not carefully tuned**, but set to commonly used values. As shown in **Figure 3**, the reward curve is naturally symmetric and stable.
> * **Stage 3** applies a length-aware shaping strategy using **default (α, β)** values adopted directly from GRPO-LEAD.
>
> Thus, while the overall pipeline appears multi-stage, each stage was deliberately designed with **minimal tuning and clear interpretability**.
>
> Looking forward, we believe it is possible to **unify these stages through a more holistic reward formulation**, enabling the model to learn adaptive reasoning behavior in a single-stage process. This is an exciting direction we intend to pursue in future work.
>
> ---
>
> If you have any additional feedback or suggestions, we would greatly welcome them and be happy to address them in detail.

---

> > ### Comment · Reviewer_x9j5 · 2025-08-04
> >
> > Thank you for the response and explanations.

---

> > > ### Author Response · Authors · 2025-08-04
> > >
> > > Thank you again. We appreciate the time and effort you took to engage with our work.

---

### Official Review · Reviewer_9eSD · 2025-07-02

**Clarity:** 2
**Significance:** 2
**Originality:** 2
**Rating:** 3
**Confidence:** 4

**Summary:**

This paper introduces AutoThink, a framework designed to mitigate the over-thinking problem in large reasoning models, where models expend excessive computational resources on simple problems. The authors discover that a minimal "ellipsis prompt" (`...`) can stochastically induce R1-style models to either engage in detailed reasoning ("think" mode) or provide a direct answer ("no-think" mode). To harness this behavior, they propose a three-stage RL process.

**Questions:**

- How to define Thinking and No-Thinking? I did not find precise definition of this basic concept. Is it a statistic by counting the response length? This is the foundation of the work. Please provide clear definition.

**Ethical Concerns:**

["NO or VERY MINOR ethics concerns only"]

**Final Justification:**

My major concern of this paper, the complexity and generalization ability of the proposed method, still remain unsolved. It would be hard for the academic and open-source community to learn from this work given its complixity. Thus, I maintain my boarderline rejection rating.

**Limitations:**

Yes

**Quality:**

3

**Strengths And Weaknesses:**

Strengths
- The discovery of the "ellipsis prompt" is a interesting. It is a simple, low-cost method to unlock latent adaptive behavior in existing R1-style models without complex architectural changes
- The investigated overthinking problem is intime and important.
- Writing is clear and fluent, easy to follow

Weakness
- The proposed Elipsis prompt seems as an intermieda point between thinking and no-thinking prompts, and only shows effect on DeepSeek-R1-Distill-Qwen models. The claim that it triggers selective reasoning is not validated on broader models and scenarios.
- The 3-stage training recipe is similar to DeepScaleR. However, the resulting model starting from DS-distill-Qwen-1.5B underperforms DeepScaleR (51.7% vs 56.7% acc, 5108 vs 5817 length)
- There are too many design choices in the proposed method, including prompt selection, 3-stage training, hyperparameters in reward design. It is hard to tell which part is essential. The ablation study is simple

---

> ### Author Rebuttal · Authors · 2025-07-30
>
> We thanks for your thoughtful feedback and valuable suggestions. We have carefully addressed each concern and added clarifications, analyses, and new results to strengthen our work.
>
> ---
>
> **1. Definition of No-Thinking**
>
> Thank you for pointing this out. In R1-style models (e.g., DeepSeek-R1), *Thinking* refers to generating a step-by-step reasoning trace within the `<think> {thinking_content} </think>` tag, typically involving continuous reflection and backtracking. This structured reasoning format has been adopted in recent large reasoning models with explicit `<think>`-style reasoning blocks to modularize thought and improve interpretability.
>
> For the *No-Thinking* mode [1],  the model **immediately closing the `<think>` tag without producing any meaningful intermediate reasoning**, e.g., generating a shallow structure like `<think> </think>` and proceeding directly to the final answer. **It is also called as *Non-Thinking* mode in Qwen3 Technical Report.** This phenomenon often emerges under our **ellipsis prompt**, which stochastically toggles the model between reasoning and shortcut modes. Our work leverages this property as a *control signal* to investigate adaptive reasoning behavior.
>
> We apologize for not formalizing this definition clearly in the original submission and will include more detailed definitions, examples, and connections to related work in the camera-ready version.
>
> *[1] Reasoning models can be effective without thinking. arXiv:2504.09858.*
>
> ---
>
> **2. Validation beyond DeepSeek-R1-Distill-Qwen**
>
> Thank you for this valuable comment. We would like to clarify that our method specifically targets at **Large Reasoning Models**. Among open-source models, **DeepSeek-R1-Distill-Qwen** is a representative R1-style distilled model trained from Qwen2.5 by using DeepSeek-R1-671B reasoning data, making it a natural choice for our main experiments.
>
> To further validate generality, we have already included **DeepScaleR** in our experiments (Table 1), which is built upon DeepSeek-R1-Distill-Qwen-1.5B and strengthened via multi-stage context extension and RL. Even though DeepScaleR has undergone extensive fine-tuning to favor long reasoning chains, we observe that under the **ellipsis prompt**, the average token usage still drops from 5817 to 5511, suggesting that our prompt design still elicits a small amount of no-thinking behavior. Moreover, **AutoThink stages applied on top of DeepScaleR** demonstrate clear ability to further reduce token usage with minimal or no accuracy loss (see Table 1).
>
> In addition, we conducted further validation on **Skywork-OR1-Math-7B**, a stronger RL-trained R1-style model that has undergone deeper optimization for mathematical and code reasoning. We report below the results on different prompting/training stages:
>
> |Skywork-OR1-Math-7B|Accuracy (%)|Token Usage|
> |-|-|-|
> |Standard prompt|70.5|9053|
> |Ellipsis Prompt|70.9|8933|
> |No-Thinking Prompt|54.7|2722|
> |AutoThink-Stage1|68.2|3966|
> |AutoThink-Stage2|69.7|4719|
> |AutoThink-Stage3|68.8|3974|
>
> Although **ellipsis prompting alone induces minimal no-thinking behavior (only ~0.5% occurrence), our method successfully encourages such behavior to emerge through structured reward shaping**, reducing the average token usage from 8933 to 3966 in Stage 1 while maintaining comparable accuracy. During Stage-1 training, we observed a sharp drop in thinking ratio and response length around step 120, indicating a successful emergence of dual-mode balance.
>
> Finally, due to time and compute constraints during the rebuttal phase, we were unable to run the full training pipeline on **Qwen3**. However, we conducted a preliminary evaluation to assess the effectiveness of ellipsis prompting, recording both accuracy and token length:
>
> |Qwen3-8B|MATH500|Minerva|Olympaid|AIME24|AMC23|Avg|
> |-|-|-|-|-|-|-|
> |Standard prompt|97.0/5351|53.3/7010|73.5/11342|86.7/14690|88.1/10343|79.7/9747|
> |Ellipsis Prompt|96.4/5109|49.5/5315|70.9/9891|68.3/13349|88.9/9858|74.8/8704|
> |No-Thinking Prompt|84.1/1104|41.2/639|50.8/2860|26.3/6518|60.1/2913|52.5/2807|
> |Ellipsis Prompt: Thinking Rate|96.9%|67.5%|89.0%|87.5%|96.2%|87.4%|
>
> The use of the ellipsis prompt can encourage a certain degree of no-thinking behavior, and this behavior does not correlate with task difficulty (AIME problems are significantly harder than those in MATH500, yet elicit a lower thinking rate). **Based on the experimental results from the above Skywork-OR1-Math-7B (about 0.5% occurrence of no-thinking behavior with Ellipsis Prompt), we believe that the AutoThink strategy could also induce autonomous reasoning in Qwen3 (~13% occurrence of no-thinking behavior with Ellipsis Prompt)**.
>
> ---
>
> **3. Why AutoThink underperforms DeepScaleR despite similar training recipe?**
>
> This is a reasonable concern. We emphasize that **DeepScaleR and AutoThink are designed with different optimization objectives**:
>
> * DeepScaleR focuses solely on **maximizing accuracy**, leveraging a deep 3-stage RL training schedule with progressively extended context and a large computational budget (~2200 H100 GPU hours).
> * In contrast, **AutoThink is designed to trade off accuracy and efficiency**, aiming to **reduce unnecessary reasoning** by learning when to think, just costs ~700 H100 GPU hours.
>
> More importantly, we also **apply AutoThink on top of DeepScaleR**. As shown in Table 1, this leads to:
>
> * A **reduction in average token usage from 5817 to 5277**,
> * While **maintaining comparable accuracy** (from 56.7% → 57.3%).
>
> This shows that our method can serve as a **post-training compression module**, applicable even to models like DeepScaleR that are already highly optimized. In principle, for excellent performance and efficiency, we can first train a model with the DeepScaleR recipe to maximize performance, and then apply AutoThink to further improve inference-time efficiency without significant performance drop.
>
> ---
>
> **4. Concern about too many design choices and simple ablation**
>
> We appreciate your concern regarding the complexity of our method. We would like to clarify that the **three-stage structure is intentionally modular and interpretable**, with each component driven by distinct and simple design principles:
>
> * **Stage 1** introduces a batch-wise reward balance to prevent mode collapse between thinking and no-thinking behaviors.
> * **Stage 2** focuses purely on reinforcing accuracy within each mode (no additional reward shaping).
> * **Stage 3** adds length-aware shaping to encourage brevity for correct responses and elaboration for incorrect ones.
>
> Among the three stages, only Stage 1 and Stage 3 involve reward design beyond naive correctness. Even then, the reward formulations are simple and intuitive:
>
> * In **Stage 1**, we balance the modal ratio using a linear penalty with hyperparameters γ and λ. These parameters were not carefully tuned, but simply set to commonly used values. As shown in Figure 3, the reward curve exhibits a natural symmetric form.
> * In **Stage 3**, we reuse default shaping terms (α, β) inspired by GRPO-LEAD, additional any revision.
>
> We have already provided **ablations in Section 4.3 and Figure 5**, which show:
>
> * Without Stage 1, the model collapses into a single mode (thinking or no-thinking).
> * Skipping Stage 2 results in a clear drop in final accuracy (51.7% → 47.6%).
>
> We acknowledge the lack of hyperparameter sweeps in the main paper. In additional experiments during the rebuttal phase, we find:
>
> * Increasing **γ** encourages more thinking behavior in Stage 1.
> * Higher **λ** enforces stricter adherence to the target balance.
> * Larger **α/β** in Stage 3 accelerate reward decay/growth in response_length.
>
> To illustrate these trends, we record the *thinking ratio* measured as the running average over the most recent 20 steps at two key checkpoints (step 100 and step 200) during Stage 1 training:
>
> ||γ=0.5, λ=2|γ=0.2, λ=2|γ=0.8, λ=2|γ=0.5, λ=1|γ=0.5, λ=4|
> |-|-|-|-|-|-|
> |Thinking-Rate@step100|62.4|57.8|99.9|71.4|51.7|
> |Thinking-Rate@step200|54.2|51.4|100.0|61.2|48.3|
>
> Additionally, we record the response lengths in Stage 3 under different values of α, with β fixed at 0.05:
>
> ||α=0.05|α=0|α=0.1|
> |-|-|-|-|
> |Response-Length@step100|4734|6174|3623|
> |Response-Length@step200|4120|6322|2894|
>
> These training trends are consistent with the intended behavior of the shaping functions. We will include these findings in the appendix and release them alongside the camera-ready version.
>
> ---
>
> We sincerely hope that these clarifications and additional results help address your concerns. We would be grateful if you would kindly consider reassessing our work in light of the new evidence presented.

---

> > ### Comment · Reviewer_9eSD · 2025-08-04
> >
> > Thanks for the response. I will maintain my original rating

---

> ### Author Response · Authors · 2025-08-04
>
> Thank you for your time and thoughtful review of our response.
>
> ---
>
> Regarding the concern about the Ellipsis prompt, in addition to DeepSeek-R1-Distill-Qwen, we validated the effectiveness of the Ellipsis prompt and three-stage RL across broader model families, including Skywork-OR1-Math and Qwen3-8B.
>
> For boarder scenarios,  we conducted additional evaluations of our AutoThink-trained models on three non-mathematical benchmarks (GPQA, MMLU, LiveCodebench). The results show that AutoThink-trained models **retain competitive accuracy while reducing average token usage**, suggesting that the **learned adaptive reasoning behaviors generalize beyond math**.
>
> ||GPQA|MMLU|LiveCodeBench|Avg|
> |-|-|-|-|-|
> |DeepSeek-R1-Distill-1.5B|35.1/10026|49.5/2727|25.2/13372|36.6/8708|
> |AutoThink-Stage1|31.5/8889|47.7/1190|23.8/5653|34.3/5244|
> |AutoThink-Stage2|37.1/8617|48.8/1743|24.2/9647|36.7/6669|
> |AutoThink-Stage3|35.7/5659|48.8/1300|24.9/9054|36.5/5337|
>
> ---
>
> According to your suggestions, we analyzed the prompt selection (Standard prompt, Ellipsis Prompt and No-Thinking Prompt) , 3-stage training, hyperparameters in reward design (γ, λ and α/β). Our key findings are as follows:
>
> **Ellipsis Prompt Necessity**: The Ellipsis Prompt is critical for triggering stochastic thinking/no-thinking modes, enabling adaptive reasoning.
>
> **Training Stage Ablation**: Without Stage 1, the model collapses into a single mode (thinking or no-thinking); Skipping Stage 2 results in a clear drop in final accuracy (51.7% → 47.6%).
>
> **Hyperparameter Sensitivity**:
> - Increasing γ encourages more thinking behavior in Stage 1.
> - Higher λ enforces stricter adherence to the target balance.
> - Larger α/β in Stage 3 accelerate reward decay/growth in response_length.
>
> ---
>
> Please let us know if we have addressed your concerns. We are more than delighted to have further discussions to improve our manuscript. If our response has addressed your concerns, we would be grateful if you could re-evaluate our work.

---

> > ### Author Response · Authors · 2025-08-06
> >
> > To better validate the necessity and effect of each stage in our pipeline, we conducted additional experiments using the **Distill-1.5B model** under the **ellipsis prompting** setup. We tested the performance of the model when selectively disabling one stage at a time. The results, averaged across five math benchmarks, are summarized below:
> >
> > | Stage1 120 | Stage2 440 | Stage3 130 | Acc | Token Length| Thinking Rate |
> > |-|-|-|-|-|-|
> > | ✅ | ✅ | ✅| 51.7% | 5108 | 56.5% |
> > | | ✅ | ✅ | 51.2% | 6072 | 100% |
> > | ✅ | | ✅ | 47.6% | 6662 | 63.4% |
> > | ✅ | ✅| | 51.4% | 7295 | 42.8% |
> >
> > From these results, we observe the following:
> >
> > * **Stage 1 (balancing thinking and no-thinking)** is crucial for maintaining a moderate and efficient thinking rate. Disabling it causes the model to degenerate into always thinking (100%), leading to inefficient token usage.
> > * **Stage 2 (accuracy-driven token adaptation)** substantially contributes to final accuracy. Removing it significantly hurts performance (from 51.7% → 47.6%), even when token length increases.
> > * **Stage 3 (length regularization)** effectively trims unnecessary reasoning on easy problems, reducing token usage (5108 vs. 7295), while increasing the **thinking rate** on hard examples, highlighting its ability to **encourage deeper reasoning where needed**.
> >
> > We hope these additional results help clarify the role of each stage in our framework and demonstrate the necessity of the full design.

---

### Official Review · Reviewer_NACH · 2025-07-05

**Clarity:** 3
**Significance:** 3
**Originality:** 3
**Rating:** 4
**Confidence:** 4

**Summary:**

This work tackles the over-thinking problem in R1-style reasoning models by introducing autothink. The author develops a three-stage GRPO-based RL pipeline, to balance the thinking/non-thinking mode, refine the correct behavior, and reduce the redundant reasoning with length-aware rewards. Autothink achieves powerful performance in achieving the balance between the accuracy and the reasoning length.

**Questions:**

As discussed in weakness,

**Ethical Concerns:**

["NO or VERY MINOR ethics concerns only"]

**Final Justification:**

After carefully checking the author-reviewers' discussion, I believe most of my concerns have been resolved. And I will keep my original score.

**Limitations:**

Yes

**Quality:**

3

**Strengths And Weaknesses:**

# Strengths
1. This work owns the clear problem motivation and neat prompt insight, which shows that a mere "..." can stochastically toggle reasoning, exposing controllability overlooked by prior work.
2. A well designed multi stage RL, stage-wise reward shaping elegantly prevents mode collapse, boosts accuracy, and then trims length.
3. The proposed method achieves the best accuracy efficiency trade-off on five standard math sets and delivers significant improvement over baselines.
4. The plug and play applicability method works on both distilled and RL, suggesting broad utility of this idea.

# Weaknesses
1. The current evaluation is limited to mathematical reasoning; transfer to commonsense, coding, or general reasoning is untested.
2. The author discussed that reward - hacking (reasoning after </think>), lack of global token - budget control, and dataset noise remain open issues. Discussing how to alleviate these issues will strengthen this work.
3. Also, the training cost is only briefly compared; clearer GPU day(hours) totals compared to baselines would aid reproducibility.
4. The multi-stage will introduce extra efforts for engineering. Is it possible to build a one-stage RL by incorporating the discussed idea into it?

---

> ### Author Rebuttal · Authors · 2025-07-30
>
> We sincerely thank you for the encouraging and constructive feedback. We are glad that you found the motivation, prompt insight, and stage-wise design of AutoThink to be elegant and effective. We address your concerns below:
>
> ---
>
> **1. Generality Beyond Mathematical Reasoning**
>
> We appreciate the your concern regarding generalization beyond math tasks. In response, we have conducted additional evaluations of our AutoThink models on three non-mathematical benchmarks:
>
> * **GPQA** for scientific multi-hop reasoning,
> * **MMLU** for general multi-task language understanding,
> * **Live-Code-Bench** for code generation, we use the newest 20250727 version.
>
> The results show that AutoThink-trained models **retain competitive accuracy** while reducing average token usage, suggesting that the learned **adaptive reasoning behaviors generalize beyond math**.
>
> ||GPQA|MMLU|LiveCodeBench|Avg|
> |-|-|-|-|-|
> |DeepSeek-R1-Distill-1.5B|35.1/10026|49.5/2727|25.2/13372|36.6/8708|
> |AutoThink-Stage1|31.5/8889|47.7/1190|23.8/5653|34.3/5244|
> |AutoThink-Stage2|37.1/8617|48.8/1743|24.2/9647|36.7/6669|
> |AutoThink-Stage3|35.7/5659|48.8/1300|24.9/9054|36.5/5337|
>
> Although these findings are preliminary, they suggest that math-domain training can endow LLMs with transferable control over reasoning depth. In future work, we will explore domain-specific training to further strengthen AutoThink’s adaptability in broader tasks.
>
> ---
>
> **2. Reward Hacking, Token Budget, and Dataset Noise**
>
> We appreciate the your encouragement to further discuss the limitations we acknowledged. We agree that all three are non-trivial and active challenges in the area of reasoning control:
>
> * **Token-budget control** has been partially addressed in prior works [1,2] by designing **budget-aware reward functions** that penalize long completions. Such formulations could be integrated with AutoThink to enforce a global compute constraint in a plug-and-play manner.
>
> * For **dataset noise**, existing works ([3], etc.) shows that **curriculum learning or filtering by answer correctness or difficulty** can help improve learning efficiency. These strategies are orthogonal to our reward design and could further boost training quality when combined.
>
> * As for **reward hacking**, where reasoning continues after the `</think>` tag. we believe this can be mitigated by **explicitly penalizing reasoning-related patterns outside the `<think>` span**, or by **rewarding clean separation between thought and answer**. These ideas can be incorporated into future iterations of our reward function.
>
> Overall, we view these solutions as **complementary and composable** with our framework. We are excited to explore tighter integration of these mechanisms, and incorporate a more detailed discussion of these directions in the camera-ready version to enhance the completeness and forward-looking perspective of the paper.
>
>
> *[1] L1: Controlling How Long A Reasoning Model Thinks With Reinforcement Learning. COLM 2025.*
>
> *[2] ShorterBetter: Guiding Reasoning Models to Find Optimal Inference Length for Efficient Reasoning. arXiv:2504.21370.*
>
> *[3] FastCuRL: Curriculum Reinforcement Learning with Stage-wise Context Scaling for Efficient Training R1-like Reasoning Models[J]. arXiv:2503.17287.*
>
> ---
>
> **3. Training Cost Comparison**
>
> We thank the reviewer for the suggestion to clarify training efficiency. While we compare training steps in **Appendix B.2**, we now provide estimated **GPU-hour costs** for all methods using Distill-R1-1.5B as the base model:
>
> |Method|GPU Hours|Avg ACC/Token Length|
> |-|-|-|
> |ThinkPrune-iter-2K|\~400 H100 Hours|49.2/3368|
> |ShorterBetter|\~200 H100 Hours|44.7/1915|
> |**Distill-1.5B-AutoThink**|\~700 H100 Hours|51.7/5108|
> |DeepScaleR|\~2200 H100 Hours|56.7/5817|
> |**DeepScaleR-1.5B-AutoThink**|\~250 H100 Hours|57.3/5277|
>
> As shown, **AutoThink operates within the same order of compute as concise baselines like ThinkPrune and ShorterBetter, yet it achieves notably stronger performance.** In contrast, DeepScaleR, a performance-maximizing method, requires **over 3× more compute**, with significantly larger context length and RL iterations.
>
> We also note that **AutoThink training can be completed within 24 hours** on a 4-node H100 cluster, making it **practical and reproducible** under typical academic compute environments of R1-Style model training. **Our code is fully open-sourced, and all results in the paper are reproducible**, further supporting the accessibility and reliability of our approach.
>
> ---
>
> **4. On Single-Stage Design**
>
> We appreciate your thoughtful question regarding the complexity of our multi-stage RL framework. While AutoThink adopts a stage-wise structure for clarity and training stability, we emphasize that **each stage serves a distinct and interpretable purpose**:
>
> * **Stage 1** balances dual-mode behavior via a simple linear penalty using (γ, λ), which were **not carefully tuned** but chosen as common defaults. The resulting reward curve (Figure 3) is symmetric and robust.
> * **Stage 3** prunes redundant reasoning with length-aware shaping using **default parameters (α, β)** from GRPO-LEAD, also without fine-tuning.
>
> Thus, while multi-stage, the design is lightweight and requires **minimal hyperparameter tuning**.
>
> We agree that consolidating into a single-stage RL framework is a promising direction. One potential solution is to **dynamically switch the reward formulation during training** based on simple training signals:
>
> * When the *thinking rate* stabilizes, automatically transition from Stage 1 to Stage 2.
> * When *accuracy gain plateaus*, switch to Stage 3 to prioritize efficiency.
>
> We plan to explore such **self-adaptive reward scheduling** mechanisms in future work to reduce engineering complexity while preserving adaptive reasoning capabilities.
>
> ---
>
> We sincerely hope that these clarifications and additional results help address your concerns. If you have any additional feedback or suggestions, we would greatly welcome them and be happy to address them in detail.

---

> > ### Comment · Reviewer_NACH · 2025-08-07
> >
> > Thanks for your response. I think the rebuttal has addressed most of my concerns, and I hope the related material will be incorporated in the final version.

---

> > > ### Author Response · Authors · 2025-08-07
> > >
> > > We are grateful for your valuable feedback and the time you took to engage with our work. We will revise the related material  in the final version. If our response has addressed your concerns, we would be grateful if you could re-evaluate our work.
> > >
> > > Once again, we sincerely thank you for your valuable insights to strength our work.

---

### Comment · Area_Chair_8ES1 · 2025-08-04
**Gentle Reminder: Please Reply to Authors’ Responses (Only if Not Yet Done)**

Dear Reviewers,

As the discussion deadline approaches, may we kindly ask you to review the authors’ responses and post a constructive reply—unless you have already done so, in which case please kindly disregard this gentle reminder.

Your thoughtful engagement is deeply appreciated and essential to a fair and timely process. With sincere thanks for your continued dedication.

Area Chair

---

### Note · Authors · 2025-08-14

**Final Remark**

During the rebuttal phase, we conducted extensive additional experiments, clarified definitions, and analyzed design choices to address all reviewers’ concerns.

1. **Broader Generalization** : We evaluated AutoThink beyond mathematical reasoning on *GPQA*, *MMLU*, and *LiveCodeBench*, showing that adaptive reasoning behaviors transfer across domains while preserving accuracy and reducing token usage. We also validated on additional R1-style models, including **DeepScaleR**, **Skywork-OR1-Math-7B**, and preliminary tests on **Qwen3-8B**, confirming that ellipsis prompting and our three-stage RL remain effective even for models optimized for long reasoning chains.

2. **Clarified Definitions & Mechanisms** : We formalized the notions of *Thinking* and *No-Thinking* modes in R1-style models, and explained how the ellipsis prompt stochastically toggles between them, serving as a controllable signal for adaptive reasoning. We detailed each stage’s role: Stage 1 prevents mode collapse, Stage 2 reinforces accuracy, and Stage 3 prunes unnecessary reasoning while deepening thought on harder tasks.

3. **Ablation & Sensitivity Analyses** : We performed stage-wise ablations under ellipsis prompting, confirming that removing Stage 1 leads to 100% thinking rate, removing Stage 2 significantly lowers accuracy, and removing Stage 3 increases unnecessary token usage. We also explored hyperparameter effects (γ, λ, α/β) on thinking rate and response length, showing trends consistent with our design intentions.

4. **Efficiency & Applicability** : We compared GPU-hour costs with strong baselines, showing that AutoThink matches or exceeds their performance–efficiency trade-off while being trainable in 24 h on a 4-node H100 cluster. Applying AutoThink post-hoc to DeepScaleR reduced average token usage without harming accuracy, demonstrating modularity and compatibility with other training recipes.

5. **Limitations & Future Work** : We discussed potential mitigations for reward hacking, dataset noise, and global token-budget control, and outlined a possible unified single-stage RL via adaptive reward scheduling to reduce engineering complexity.

We believe these clarifications, results, and design analyses comprehensively resolve prior doubts, reinforcing AutoThink’s validity, generality, and practicality. We respectfully invite the AC and reviewers to consider this strengthened evidence in their final deliberations.

---

### Decision · Program_Chairs · 2025-09-17

**Decision:**

Accept (poster)

**Comment:**

Summary of the paper: This paper claims that an ellipsis prompt (“…”) inserted in R1-style models can stochastically toggle between a “think” (CoT) and a “no-think” (direct-answer) mode, providing a simple control signal for adaptive reasoning. Building on this observation, the authors design AutoThink, a three-stage GRPO-based RL pipeline: Stage 1 stabilizes coexistence of the two modes to avoid policy collapse, Stage 2 reinforces correctness inside each mode, and Stage 3 employs length-aware rewards to prune redundant reasoning. Across five math benchmarks, AutoThink yields the best reported accuracy-efficiency trade-off, cutting average response length by up to 52 % while matching or surpassing baselines in accuracy; these gains transfer to both distilled and RL-trained R1-style models. Ablations confirm that every stage is necessary and that the method mitigates mode-collapse, reward hacking, and dataset noise, although global token-budget control and broader-domain transfer remain open questions.

Strengths of the paper:
1. Simple but interesting discovery: an ellipsis prompt alone can stochastically switch reasoning modes in R1-style models, offering a low-cost, plug-and-play control handle.
2. Straightforward design: the proposed three-stage RL design with progressive reward shaping that systematically prevents collapse, improves accuracy, and trims length; ablations validate each stage.
3. Strong empirical results: achieves sota accuracy-efficiency on five math benchmarks, with substantial token savings and consistent gains across model sizes.
4. Presentation: fluent writing and extensive ablations make the contribution transparent and reproducible.

Weaknesses of the paper: After the rebuttal, I think the authors have addressed most of the concerns raised by three reviewers. Reviewer 9esd still has major concerns during the area chair-reviewers discussion phase:
1. Validity and depth: the proposed methods lack technical and theoretical depth. The ellipsis prompts, reward design in RL and multi-stage pipelines are mostly heuristics.
2. Improvement over existing work: there are already many works exploring controlling response lengths via prompting (cot, pot), RL (L1, ThinkPrune), and multi-stage training (DeepScaleR).
3. Overdesigned pipeline. AutoThink intruduced a three-staged training pipeline, which is too complicated for the community to apply. Moreover, there are too many hyperparameters to tune under this framework. The training steps across runs also varies, and the author did not propose any method to determine them. From this persipective, currecnt experimental results, which are limited on DeepSeek distilled Qwen models, do not sufficiently support their claims.

Reasons for the decision: Reviewer 9esd raises valid concerns, and I concur with their assessment of the paper’s technical depth and theoretical contribution. It seems to me that the authors addressed the hyper-parameter-tuning issue by adding a focused analysis of design choices along with a concise ablation study. They have also expanded the experimental scope to include Skywork-OR1-Math-7B and Qwen-3-8B. These revisions meaningfully strengthen the work, and—after weighing the recommendations of the other three reviewers—I am inclined to accept the paper.